# Allosteric mechanism of the *V. vulnificus* adenine riboswitch resolved by four-dimensional chemical mapping

**Siqi Tian[1], Wipapat Kladwang[1], Rhiju Das[2]\***

[1]Department of Biochemistry, Stanford University, Stanford, United States;
[2]Department of Physics, Stanford University, Stanford, United States

**Abstract** The structural interconversions that mediate the gene regulatory functions of RNA molecules may be different from classic models of allostery, but the relevant structural correlations have remained elusive in even intensively studied systems. Here, we present a four-dimensional expansion of chemical mapping called lock-mutate-map-rescue (LM$^2$R), which integrates multiple layers of mutation with nucleotide-resolution chemical mapping. This technique resolves the core mechanism of the adenine-responsive *V. vulnificus add* riboswitch, a paradigmatic system for which both Monod-Wyman-Changeux (MWC) conformational selection models and non-MWC alternatives have been proposed. To discriminate amongst these models, we locked each functionally important helix through designed mutations and assessed formation or depletion of other helices via compensatory rescue evaluated by chemical mapping. These LM$^2$R measurements give strong support to the pre-existing correlations predicted by MWC models, disfavor alternative models, and suggest additional structural heterogeneities that may be general across ligand-free riboswitches.
DOI: https://doi.org/10.7554/eLife.29602.001

## Introduction

Conformational changes in RNA molecules are ubiquitous features of gene regulation in all living cells. Recent years have seen an explosion of discoveries of *cis*-acting mRNA elements that sense small molecules, proteins, RNAs, and other environmental conditions and then modulate transcriptional termination, ribosome recruitment, splicing, and other genetic events (*Henkin, 2008*; *Nudler and Mironov, 2004*; *Gesteland et al., 2006*). Beginning with classic work by Yanofsky and colleagues in the 1970s, these discoveries have been associated with elegant models of allostery in which each molecule interconverts between distinct RNA secondary structures (*Yanofsky, 2000*; *Gutiérrez-Preciado et al., 2009*; *Monod et al., 1965*). If these multiple states could be characterized in detail and quantitatively modeled, they would offer potential new substrates for biological control and new targets for antibiotic development (*Jones and Ferré-D'Amaré, 2017*).

Despite this strong interest in RNA conformational change, directly testing the base pair correlations posited to underlie RNA allostery have been difficult for most systems. Experimental barriers remain even for the compact and self-contained riboswitches, which occur in eukaryotes and appear pervasive throughout bacteria. A riboswitch is a *cis*-regulatory RNA domain composed of two segments: (1) an 'aptamer' segment that forms an intricate tertiary structure to specifically bind small molecule ligands, such as nucleobases or amino acids, and (2) a 'gene expression platform' segment that interacts with other molecular machines like the ribosome to modulate gene expression (*Fernández-Luna and Miranda-Ríos, 2008*; *Gilbert and Batey, 2006*; *Breaker, 2012*). In many cases, the coupling of ligand binding to changes in the gene expression platform has been reconstituted in vitro in small model RNAs, enabling detailed biophysical analysis. Nevertheless, in most of these

**\*For correspondence:**
rhiju@stanford.edu

**Competing interests:** The authors declare that no competing interests exist.

cases, the mechanism underlying the coupling of ligand binding to gene expression remains under question, suggesting the need for new concepts and new biophysical methods to understand RNA allostery.

A particularly well-studied – but still incompletely understood – example of an allosteric RNA is the *add* riboswitch, a ~120 nucleotide element that resides in the 5′ untranslated region (UTR) of the *add* adenosine deaminase mRNA of human pathogenic bacterium *Vibrio vulnificus*. This riboswitch can directly bind adenine and controls translation of the mRNA in response to adenine in vivo. Domains of this RNA have been subjected to nearly every biochemical and biophysical technique available, including crystallography, single molecule measurements, and time-resolved X-ray laser diffraction (*Lemay et al., 2011*; *Mandal and Breaker, 2004*; *Batey, 2012*; *Cordero and Das, 2015*; *Stoddard et al., 2013*; *Rieder et al., 2007*; *Neupane et al., 2011*; *Lemay et al., 2006*; *Greenleaf et al., 2008*; *Wickiser et al., 2005*; *Lemay and Lafontaine, 2007*; *Stagno et al., 2017*). Almost all work on the *add* riboswitch to date have assumed some variation of a Monod-Wyman-Changeux model (MWC, also called 'conformational selection' or 'population shift' models) (*Monod et al., 1965*; *Batey, 2012*; *Gunasekaran et al., 2004*; *Changeux and Edelstein, 2005*). In the absence of adenine, the RNA dominantly forms a stable *apo* secondary structure whose aptamer region is in a fold incompatible with adenine binding and whose gene expression platform, a ribosome binding site and AUG start codon, is sequestered into helices (blue, *Figure 1A*). High adenine concentrations stabilize the ligand-binding secondary structure in the aptamer region and rearrange the secondary structure around the gene expression platform so that it is freed to recruit the ribosome and translation machinery (gold, *Figure 1A*). The signature prediction of the MWC model is that the *holo* RNA secondary structure should also exist in the absence of adenine, albeit as a high energy conformation within the *apo* ensemble (*Figure 1A*, middle panel; gold). In terms of the riboswitch functional domains, when the RNA occasionally fluctuates to form an aptamer secondary structure compatible with ligand binding, the gene expression platform is predicted to concomitantly open, even without the adenine ligand. This prediction for the ligand-free ensemble is analogous to the classic MWC model of oxygen binding by hemoglobin, involving interconversion between thermodynamically stable hemoglobin structures (tense and relaxed, dominating *apo* and holo, respectively), with transient sampling of the *holo* oxygen-binding structure even in the absence of oxygen. This prediction for hemoglobin and numerous other protein systems have been supported through decades of studies on protein allostery that include direct detection of transient *holo*-like states in the *apo* ensemble even in the absence of small molecules needed to fully stabilize the *holo* conformation (*Changeux and Edelstein, 2005*; *Eaton et al., 1999*).

Recently, a tour de force study highlighted an alternative to the MWC mechanism for the *add* riboswitch (*Figure 1B*). Multidimensional NMR spectroscopy, NMR relaxation studies, designed model systems, and supporting measurements from stopped-flow kinetics and calorimetry revealed detailed single-nucleotide-resolution base pairing information and intriguing temperature dependences for the complex structural ensemble of the *add* riboswitch (*Reining et al., 2013*). Although similar in some respects to previously proposed MWC models, the new model contradicted a standard MWC assumption (*Gunasekaran et al., 2004*; *Changeux and Edelstein, 2005*; *Eaton et al., 1999*): it proposed that, in the absence of adenine, riboswitches that sample the correct aptameric secondary structure do not also concomitantly open their ribosome binding sites and increase gene expression, as predicted by MWC conformational selection (*Figure 1B*, middle panel; mixed blue and gold, red rectangle outline). Despite this fundamental distinction, the non-MWC model appears fully consistent with all available data, and it remains unclear what pre-existing structural correlations, if any, underlie the allosteric mechanism of the *add* riboswitch. As an illustration of current uncertainties in the system, the same group that first proposed an non-MWC model for the *add* RNA has recently revived the MWC framework to interpret newer single molecule measurements on the same system (*Warhaut et al., 2017*). Understanding the mechanism of allostery is particularly important for developing strategies for controlling riboswitches. For example, potential antibiotics that attempt to constitutively turn on expression of the *add* gene by selectively stabilizing the aptamer secondary structure may not succeed if the MWC model is incorrect.

Here, we develop an experimental approach that resolves the allosteric mechanism of the *add* riboswitch. To help avoid previous confusion in state definitions, we first cast the question of mechanism into one of determining the statistical correlation or anticorrelation of specific helix elements. We then expand a previously reported method that integrates compensatory rescue with chemical

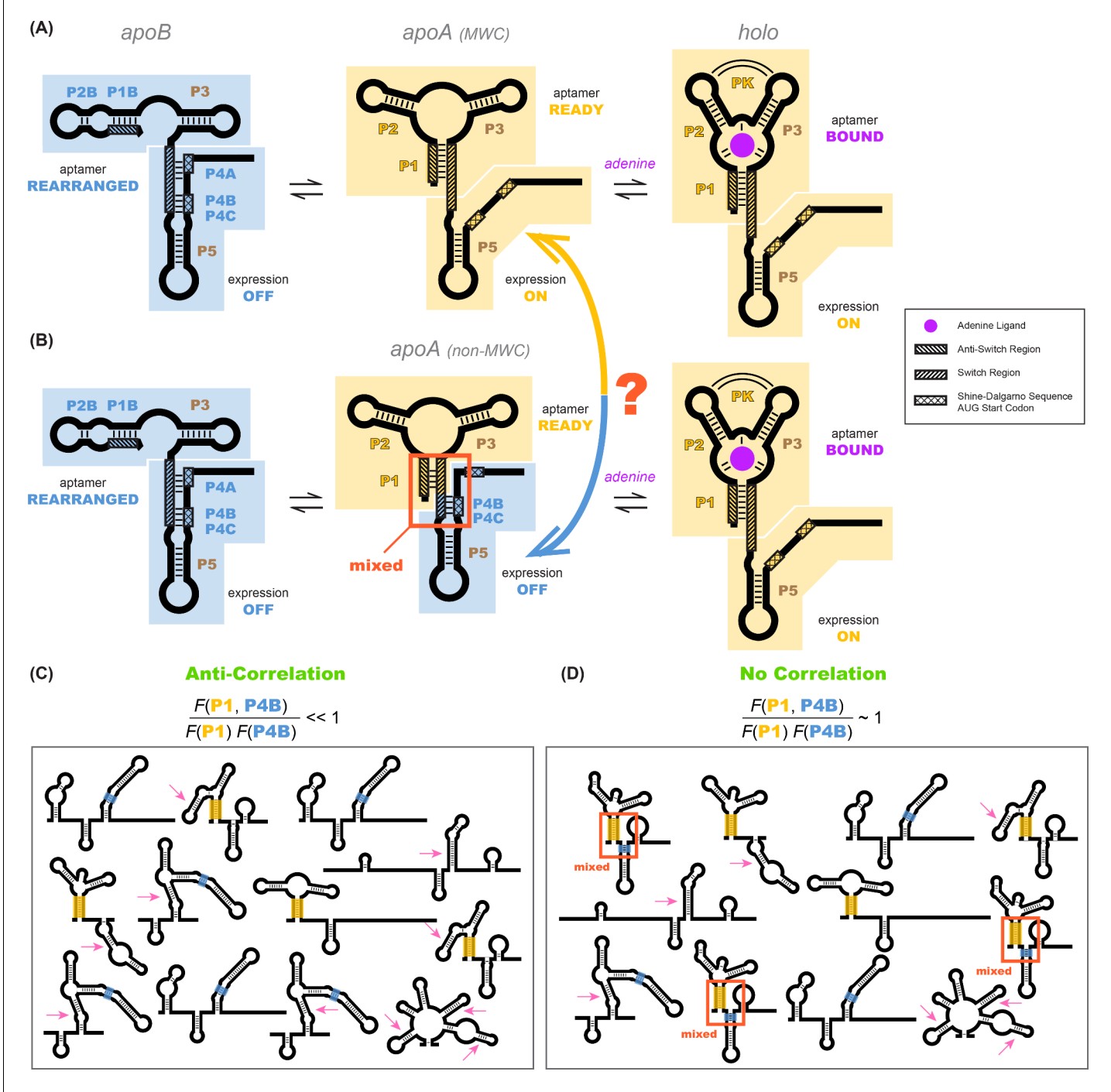

**Figure 1.** Models of the *add* riboswitch. (A–B) Three-state models for conformational changes and ligand binding in the *add* riboswitch. In current models, there is an *apoB* state (left). This state is OFF (Shine-Dalgarno sequence and AUG start codon sequestered) and is also incompetent for ligand binding due to the aptamer's rearrangement (blue background in both regions). The ligand-bound state *holo* (right) is ON with Shine-Dalgarno sequence and AUG codon (gene expression platform, cross-hatched) exposed and the aptamer region folded into a secondary structure compatible with binding adenine (gold background in both regions; adenine is purple). The *apoA* state (center), defined as the transient adenine-free conformation with an aptamer fold ready for ligand binding (gold background), differs in the two models. (A) In a Monod-Wyman-Changeux model, *apoA* is assumed to be ON, with a similar expression platform to *holo*. (B) In a non-MWC model proposed in a prior study (***Reining et al., 2013***), the *apoA* state is 'mixed', with the aptamer fold ready for binding (gold) but the expression platform turned OFF as the Shine-Dalgarno sequence and AUG codon are sequestered into helices P4B and P4C (blue). (C–D) Cartoons of possible ligand-free structural ensembles allowing for myriad alternative secondary structures with only some helices shared with those in (A–B) or completely different helices. This description does not assume a three-state

*Figure 1 continued on next page*

*Figure 1 continued*

decomposition of (**A–B**), which does not take into account the possibility of other possible helices (pink arrows). In an MWC scenario (**C**), helices characteristic of the aptamer secondary structure (such as P1, highlighted gold) and helices characteristic of closed gene expression platform (such as P4B, highlighted blue) appear in different members of the structural ensemble, but never together. The helix frequencies are anticorrelated (equation above cartoon). In non-MWC scenario (**D**), both kinds of helices appear together in 'mixed' secondary structures (marked with red arrows) at joint frequencies similar to the product of their individual occurrence frequencies (equation above cartoon).

DOI: https://doi.org/10.7554/eLife.29602.002

The following figure supplements are available for figure 1:

**Figure supplement 1.** One-dimensional SHAPE profiles of *add* riboswitch constructs.

DOI: https://doi.org/10.7554/eLife.29602.003

**Figure supplement 2.** Mutate-and-map (M$^2$) experiments on the *add* riboswitch.

DOI: https://doi.org/10.7554/eLife.29602.004

**Figure supplement 3.** Candidate alternative helices from M$^2$ analyses.

DOI: https://doi.org/10.7554/eLife.29602.005

mapping readouts (mutate-map-rescue, M$^2$R) (*Tian et al., 2014*; *Tian and Das, 2016*) to enable dissection of these helix-helix correlations. The method takes advantage of a Bayesian framework and simulations to connect experimental observables to underlying helix frequencies, and gives experimental results on the wild type *add* sequence that agree with predictions from all available models. Then, by identifying 'lock' mutants that stabilize helices posited for each state and carrying out further rounds of M$^2$R in these lock mutant backgrounds, we infer how the presence of one helix enhances or suppresses the presence of other helices. The resulting data unambiguously discriminate between MWC and non-MWC models and also suggest additional structural heterogeneity for the *add* riboswitch. We end by discussing how the presented method may be useful for generally dissecting allostery in RNA-based gene regulation.

## Results

### Defining the question in terms of helix-helix correlations

Before presenting our experimental strategy to dissect RNA allostery, we need to state the problem in a way that can be answered in a binary fashion. As shown in *Figure 1A and B*, a conventional way to phrase questions about allostery is to posit a small number of states and then to design experiments to probe properties of those states. For example, for the *add* riboswitch, previous models partitioned the states in the *apo* ensemble into *apoA* and *apoB*, depending on whether the aptamer secondary structure is correctly formed or not, respectively. We would then design experiments to discriminate whether *apoA* has its gene expression platform open or closed (*Figure 1A* vs. *Figure 1B*). However, in practice, this way of posing the question can be confusing. *Figure 1C and D* show cartoons of the ensemble of RNA conformations that illustrate the potential complexities. First, there may be RNA secondary structures present in the actual *apo* ensemble that include some but not all helices of the aptamer secondary structures (*Figure 1C and D*), and it then becomes ambiguous whether they should count as members of the *apoA* or *apoB* state. In addition, there may be secondary structures that include or are entirely composed of helices not shown in any of the presented models (marked with pink arrow, *Figure 1C and D*). Indeed, in principle, there are a vast number of secondary structures that need to be considered. Not only are there several possible helices that can be formed by the RNA, but there are an exponentially large number of combinations of such helices, each giving rise to a distinct secondary structure.

We sought to define the allosteric mechanism in terms of the possible helices that the RNA can form, rather than this vast number of possible secondary structures. In the presence of adenine, there are three helices P1, P2, and P3 that define an adenine-binding aptamer secondary structure at nucleotides 15–81, with a well-defined tertiary structure that has been determined through crystallography. A linker including an additional hairpin (P5, nts 89–110) connects the aptamer to the Shine-Dalgarno sequence and AUG start codon (nts 112–122), which are open and available for ribosome recruitment. In the absence of adenine, all available models posit that the riboswitch forms a secondary structure ensemble dominated by a new P4 domain that pairs with the Shine-Dalgarno

sequence and the AUG start codon (nts 112–122). The P4 domain includes potentially three helices, here called P4A, P4B, and P4C. P4A contains nucleotides that were originally part of P1 (the 'switch region' [*Batey, 2012*], nts 76–81, cross-hatched in *Figure 1A*, left). When P4A forms, some or all of the aptamer helices may rearrange into an alternative secondary structure or, potentially, a large set of alternative secondary structures. NMR experiments directly detected two alternative helices, P1B and P2B (*Figure 1A*), but do not rule out other helices being present at significant fractions if their conformational dynamics renders them invisible to the spectroscopy methods applied. Despite these potential complexities, there are a small number of Watson-Crick helices that feature in the current MWC and non-MWC models: P1, P2, P3, P4A, P4B, P4C, and P5. Additional helices may also be relevant to the *add* riboswitch mechanism. For example, numerous helices arose in preliminary modeling of the riboswitch based on conventional 1-dimensional SHAPE and mutate-and-map measurements (Supplementary Materials; *Figure 1—figure supplement 1*, *Figure 1—figure supplement 2*, and *Figure 1—figure supplement 3*). These helices (called here P6, P8 to P17) are further tested below.

Given the list of candidate helix elements, we can define the question that discriminates between MWC and non-MWC models, even if the *apo* ensemble is highly heterogeneous (*Figure 1C and D*). In the absence of adenine, the riboswitch may occasionally sample the helices P1 or P2 (or both simultaneously), which are signatures of the adenine-bound state. The MWC conformational selection model would assume that any such thermally sampled secondary structure should also have an open gene expression platform, just like the adenine-bound *holo* and without the P4A or P4B domains. That is, there should be an anticorrelation between the presence of, e.g., P1 and P4B, in the ligand-free ensemble (*Figure 1C*). There should be similar anticorrelations between, e.g., P2 and P4A. (For clarity of presentation, here and below, we will use P1 and P4B (colored gold and blue, respectively, in figures) to illustrate core concepts and data in detail, and then summarize additional independent measurements for other helix-helix pairs, such as P2 and P4A.)

More quantitatively, suppose that P1 and P4B arise with 50% frequency in the ligand-free *add* riboswitch structural ensemble. If there is no allosteric 'communication' between these two elements, the joint frequency of observing both P1 and P4B should be 50% x 50% = 25%. The MWC model predicts that the actual joint frequency of these two elements is significantly lower: the formation of P1 in the aptamer destabilizes the P4B helix that sequesters the gene expression platform, even in the absence of adenine. In terms of a single numerical value, we seek to determine the correlation $g(\mathrm{P1}, \mathrm{P4B})$:

$$g(\mathrm{P1}, \mathrm{P4B}) = \frac{F(\mathrm{P1}, \mathrm{P4B})}{F(\mathrm{P1}) \times F(\mathrm{P4B})} \tag{1}$$

where *F* is the helix frequency, a number between 0% and 100%.

The MWC model predicts a strong anticorrelation: $g(\mathrm{P1}, \mathrm{P4B}) \ll 1$. In contrast, the non-MWC model proposed based on NMR experiments makes different predictions (*Figure 1B and D*). In the absence of ligand, the NMR measurements could not directly detect opening of the gene expression platform (loss of P4A, P4B, or P4C) at low frequency. The non-MWC model suggests instead a mixed conformation for *apoA*, where a shortened P1, P2, and P3 form the adenine aptamer secondary structure but also some parts of the P4 domain (here denoted P4B and P4C) are retained (middle panel of *Figure 1B*, and structures marked with red arrow in *Figure 1D*). The non-MWC model predicts no correlation between elements of the aptamer region and elements of the P4 domain, so that, for example, $g(\mathrm{P1}, \mathrm{P4B}) \approx 1$.

Our strategy involves experimentally determining the helix frequencies in the *add* riboswitch structural ensemble $F(\mathrm{P1})$, $F(\mathrm{P4B})$, etc., based on a method described in the next section. To then determine the correlation for two elements P1 and P4B, we design mutations to lock P4B and see if the frequency of the P1 element increases, stays the same, or decreases. Such measurements allow inference of $g(\mathrm{P1}, \mathrm{P4B})$ through the relation of joint probabilities to conditional probabilities, $F(\mathrm{P1}, \mathrm{P4B}) = F(\mathrm{P1}|\mathrm{P4B})F(\mathrm{P4B})$. *Equation (1)* becomes

$$g(\mathrm{P1}, \mathrm{P4B}) = \frac{F(\mathrm{P1}|\mathrm{P4B})}{F(\mathrm{P1})} \tag{2}$$

Here $F(\mathrm{P1}|\mathrm{P4B})$ is the frequency of helix P1 conditional on the formation of P4B and would be

Biochemistry and Chemical Biology

estimated from experiments that lock P1 and assess the P4B helix frequency. In a 'flipped' experiment, we also lock P1 and then assess if the helix frequency of P4B increases, stays the same, or decreases. The results enable an independent evaluation of the same correlation value through a similar relation:

$$g(\text{P1}, \text{P4B}) = \frac{F(\text{P4B}|\text{P1})}{F(\text{P4B})} \tag{3}$$

Correlation values for other pairs of helices, such as $g(\text{P2}, \text{P4B})$, are defined analogously to *Equations (1)-(3)*.

## Inferring helix frequencies through quantitative compensatory rescue

To estimate correlations between helix elements, we need a way to estimate frequencies of each helix element in an RNA's structural ensemble, the variables $F(\text{P1})$, $F(\text{P4B})$, etc. in *Equations (1)-(3)*. We discovered an approach to such helix frequency inference when revisiting the use of compensatory mutation experiments for RNA structure read out through SHAPE chemical mapping (selective 2′-hydroxyl acylation with primer extension) (*Tian et al., 2014*; *Tian and Das, 2016*). Mutate-map-rescue (M²R) experiments (*Figure 2*) enable incisive confirmation or falsification of candidate base pairs. Strong evidence for a candidate base pair comes from observations that mutation of either side of a base pair (mutant A or mutant B) disrupts the SHAPE profile of the RNA; and that the compensatory double mutation (mutant AB) restores the SHAPE measurements to the wild type (WT) profile. In our prior work on a four-helix junction from the *E. coli* 16S ribosomal RNA (nts 126–235), we were able to successfully use such 'quartets' of SHAPE profiles (WT, mutant A, mutant B, mutant AB) to discriminate between three conflicting structural models derived from prior SHAPE data on the wild type RNA alone, from mutate-and-map data (SHAPE profiles of all single mutants), and from phylogenetic and crystallographic models of the protein-bound ribosome (illustrative data for three base pairs given in *Figure 2A*) (*Tian et al., 2014*). However, we were also initially surprised to observe compensatory rescue for helices that should have been mutually exclusive in the 16S rRNA. These helices involved the same nucleotides partnered to nucleotides differing by a single-nucleotide register shift (P4a and shift-P4a, not to be confused with P4A in the *add* riboswitch; see blue and green base pairs in *Figure 2A*). Resolving this paradox, studies on designed mutants and quantitative fits suggested that both registers were indeed present but in different subsets of the RNA's structural ensemble, at helix frequencies of ~20% and ~80% (*Tian et al., 2014*).

In our studies on the 16S rRNA four-way junction, the strongly populated helix shift-P4a was associated with mutate-map-rescue data that showed unambiguous rescue while the less populated helix P4a was associated with 'partial' rescue. In the latter case, disruptions in SHAPE reactivity induced by single mutations were alleviated by their concomitant compensatory mutation, but new features distinct from the wild type SHAPE profile also appeared (yellow arrow, bottom panel of *Figure 2A*). We speculated that such 'partial rescue' might generally be a hallmark of base pairs that are present at low frequency in an RNA's structural ensemble. We envisioned the structural ensembles shown in *Figure 2B*, illustrating a case where a candidate base pair is present at 50% frequency in the wild type RNA structural ensemble (structures with gold highlight rectangle in top left, *Figure 2B*). The compensatory double mutant (structures with gold highlight rectangle in bottom right, *Figure 2B*) might retain the structures with the candidate base pair at approximately 50% frequency, while the other members of the structural ensemble would be more likely to rearrange (bottom structure of bottom right panel, *Figure 2B*). The single mutants would rearrange all structures, including the ones that had the target base pair (other structures, bottom left and top right panels, *Figure 2B*), providing an assessment of the maximum level of disruption that might be restored by the compensatory mutant. In general, however, this picture is not expected to be exact. Some compensatory double mutations might over- or under- stabilize the structures with the target base pair. The RNA may also be excessively sensitive or insensitive to disruption by single mutations. Therefore, we expected that testing all the base pairs of a target helix through separate mutate-map-rescue experiments for each base pair would be critical for assessing helix frequencies, and a careful treatment of uncertainties would be important for establishing statistical confidence.

To test this idea, we first developed a quantitative 'rescue factor' metric that allowed us to avoid human inspection that might bias assessments:

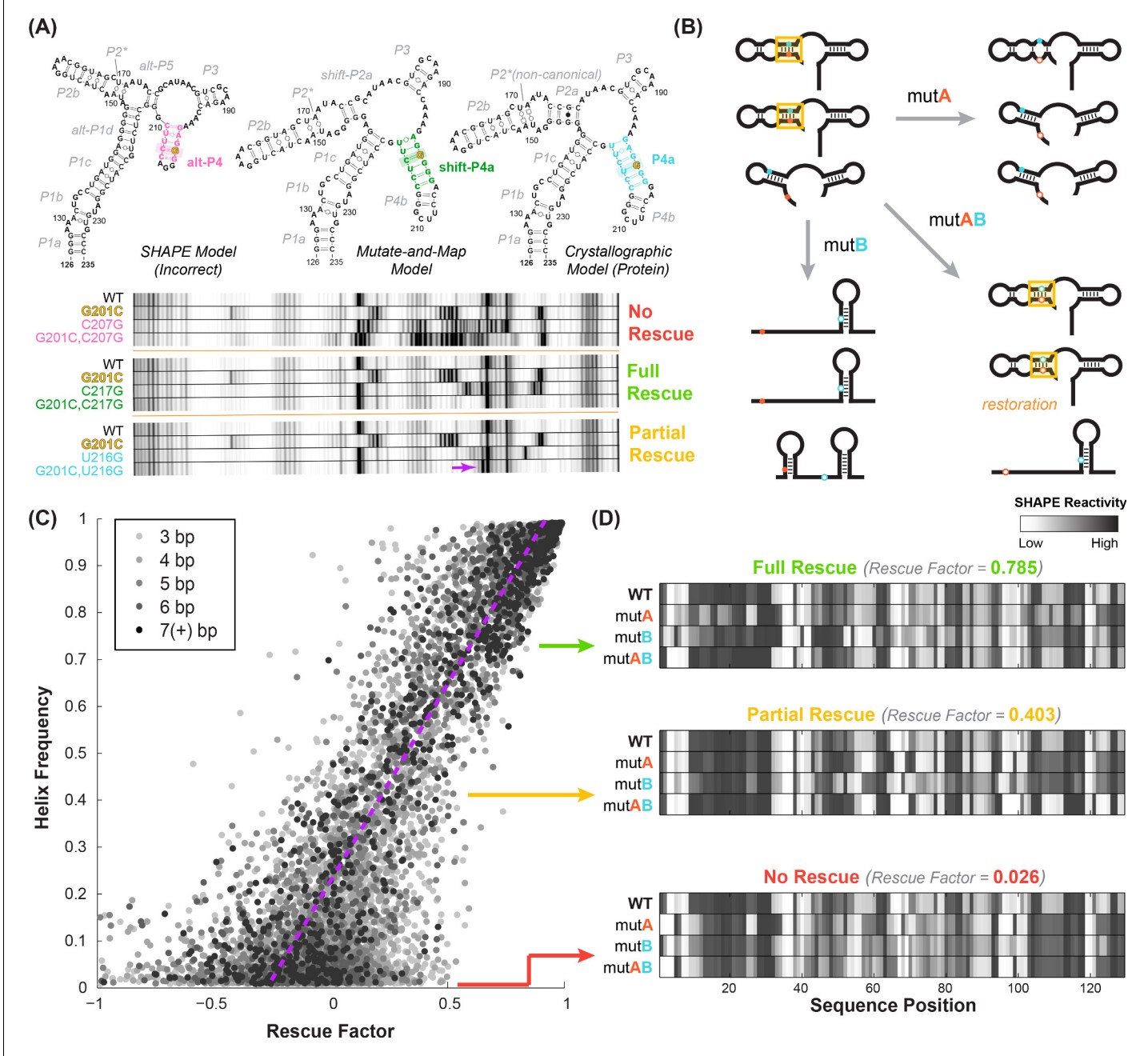

**Figure 2.** Mutate-map-rescue (M²R) enables inference of helix frequencies. (**A**) Interrogation of base pairs by compensatory mutagenesis and chemical mapping, illustrated on a four-way junction (nts 126–235) of the *E. coli* 16S ribosomal RNA. Experiments involve 'quartets' of SHAPE profiles for the wild type RNA, variants with single mutations on each side of a candidate base pair, and a variant with double mutations that could rescue Watson-Crick pairing of the candidate pair. Experiments involving possible pairings of G201 result in no obvious rescue (G201C-C207G), clear rescue (G201C-C217G), or an intermediate result (G201C-U216G), termed 'partial rescue'. In partial rescue, some disruptions observed in single mutants are rescued, but other features are not rescued or arise in the double compensatory mutant (marked with yellow arrow). Panel adapted from ref (*Tian et al., 2014*). (**B**) Cartoon of structural ensembles that explain the 'partial rescue' scenario. A fraction of structures in the wild type ensemble have target base pair (cyan/ blue, within helix outlined in gold); this picture assumes that those structures are disrupted in the single mutant but restored with similar fraction in the double mutant ensemble. (**C**) Correlation of observed rescue factor against helix frequency in simulated mutate-map-rescue experiments. Data are for helices with length longer than 2 bp for RNAs across 325 Rfam families mutated and folded in silico. Each data point represents a helix whose rescue factor and simulated helix frequency has been averaged across all its base-pairs. Helices are shaded in grays based on their length. The rescue factor estimates the extent of similarity restored in a double mutant's reactivity compared to WT (wild type), scaled by perturbations observed in single

*Figure 2 continued on next page*

*Figure 2 continued*
mutants. *(B) In silico* M²R quartets illustrating full compensatory rescue (high rescue factor), partial rescue (medium rescue factor) and no rescue (low rescue factor).
DOI: https://doi.org/10.7554/eLife.29602.006
The following figure supplement is available for figure 2:

**Figure supplement 1.** Correlation between rescue factor and helix frequency from Rfam simulations of mutate-map-rescue (M²R).
DOI: https://doi.org/10.7554/eLife.29602.007

$$\text{Rescue factor} \equiv 1.0 - \frac{RMSD(\text{WT}, \text{AB})}{\max(RMSD(\text{WT}, \text{A}), \ RMSD(\text{WT}, \text{B}))}, \tag{4}$$

where the root mean squared difference *RMSD* between two SHAPE profiles *d* over probed nucleotides *i* = 1 to *N* is given by:

$$RMSD(\text{A}, \text{B}) = \sqrt{\frac{1}{N}\sum_{i=1}^{N}[d_A(i) - d_B(i)]^2} \tag{5}$$

In *Equation (4)*, the term $RMSD(\text{WT}, \text{AB})$ captures the extent of rescue, as compared to a maximal amount of disruption of the SHAPE profile given by the maximum of $RMSD(\text{WT}, \text{A})$ and $RMSD(\text{WT}, \text{B})$. We confirmed that this rescue factor recovered expert assessments for presence or absence of rescue in the previous 16S rRNA data as well as in newly collected M²R experimental data, including the *Tetrahymena* ribozyme P4-P6 domain and a blind RNA-puzzle modeling challenge, the GIR1 lariat capping ribozyme (Materials and Methods and Appendix Results). We also developed a separate category-based classification to check that compensatory rescue of different base pairs within the same helix gave concordant results, and again that automated analysis reproduced manual analysis (Appendix Results).

We then sought to test our hypothesis of a quantitative relationship between rescue factor and helix frequency in an RNA's structural ensemble. To accumulate good statistics and ensure good comparisons to 'ground truth', we carried out simulations of M²R experiments for structural ensembles of hundreds of naturally derived RNA sequences using the *RNAstructure* simulation package *Reuter and Mathews, 2010*. These in silico tests confirmed that observation of strong rescue over all the base pairs of a helix implies high helix frequency (*Figure 2C*). Importantly, these tests also showed that modest rescue ratios could be informative. Values lower than one but greater than zero corresponded to 'partial' rescue, where the reactivities at some nucleotides revert to wild type (WT) values but others do not (*Figure 2D*, center). As expected, individual base pair tests were noisy, but averaging rescue factors over tests on three or more base pairs gave values that showed a striking correlation with helix frequencies (compare gray and black points, *Figure 2C*). Furthermore, absence of rescue (rescue factor close to zero; *Figure 2D*, bottom) gave upper bounds on the helix frequency (*Figure 2—figure supplement 1*). To estimate uncertainties in these estimates, we used a Bayesian framework to derive posterior probabilities of each helix frequency given the observed rescue factor, calibrated based on these simulations. These posterior probability distributions represent our belief, informed by experiment, at all helix frequencies from 0% to 100% and provide a complete representation of our uncertainty; 'fatter' distributions correspond to larger uncertainties in helix frequencies. For ease of reading in the main text and in summary tables, we give median values of the helix frequency posterior probability distributions. Our simulation and experimental studies suggested that compensatory rescue measurements, including ones that resulted in partial or no detectable rescue, could be used to estimate helix frequencies and uncertainties for those values, though we sought more detailed experimental tests, as described next.

## Tests of helix frequency estimation on wild type *add* riboswitch sequence

To test the applicability of M²R helix frequency estimation for understanding complex RNA structural ensembles, we first applied the method to the *add* riboswitch with and without adenine (*Figures 3* and *4*). These measurements were necessary to determine baseline values of each helix frequency –

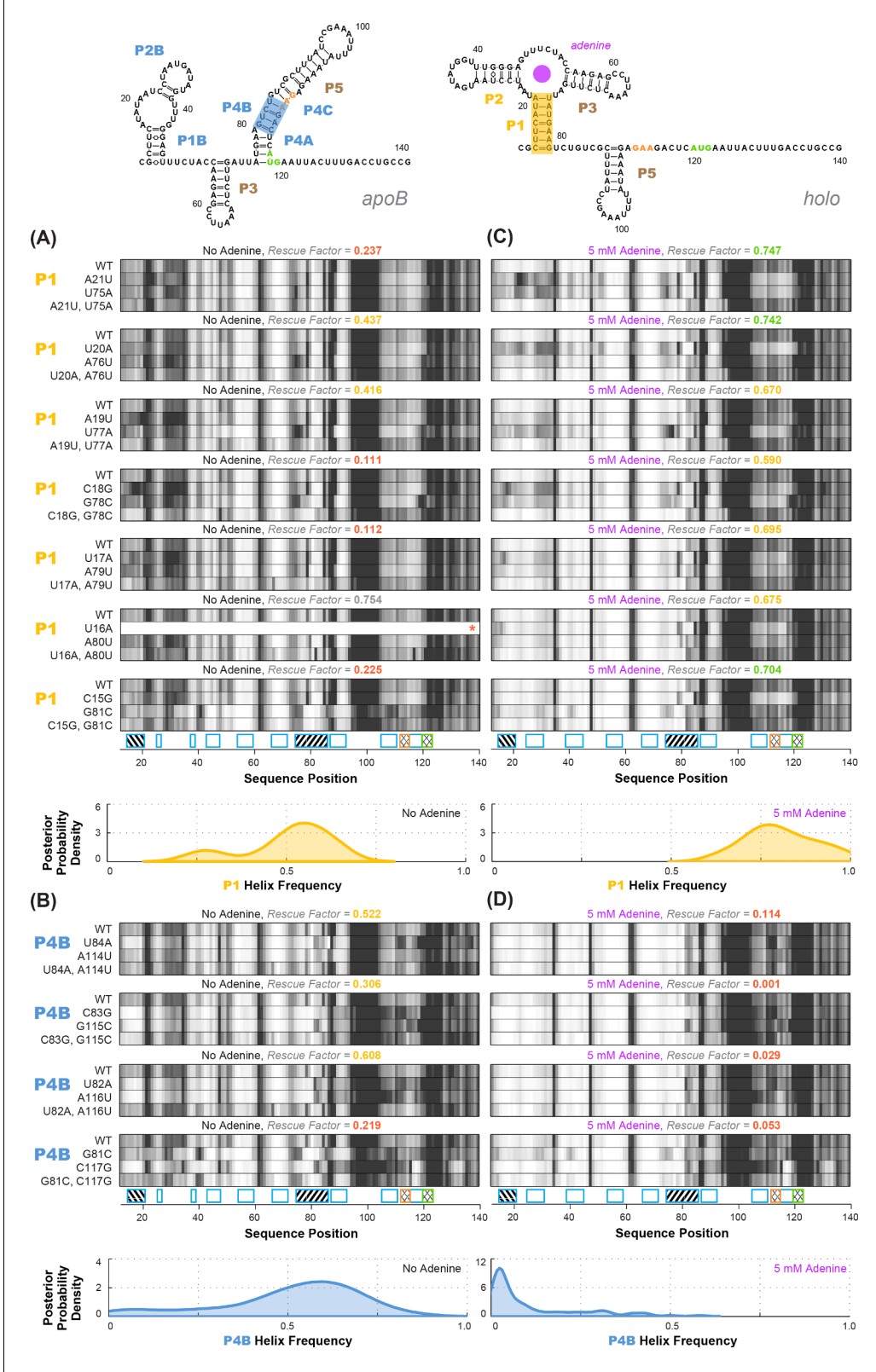

**Figure 3.** Experimental data and frequency estimates for helices P1 and P4B in the *add* riboswitch structural ensembles. Top: two example secondary structures showing sequences of the *add* riboswitch and helices probed. The presented structures are for illustration of the helices only, and do not imply co-existence in the same structure. In (A-D), each panel shows (top) experimentally measured mutate-map-rescue ($M^2R$) quartets for each base pair of a probed helix and (below) posterior distributions over helix frequency, estimated from the mean rescue factor in the $M^2R$ experimental data

*Figure 3 continued on next page*

*Figure 3 continued*

and simulations (see *Figure 2—figure supplement 1*). Experiments are shown probing (A) P1 helix without adenine, (B) P4B helix without adenine, (C) P1 helix with 5 mM adenine, and (D) P4B helix with 5 mM adenine. Measurements were acquired in 10 mM $MgCl_2$, 50 mM Na-HEPES pH 8.0.

DOI: https://doi.org/10.7554/eLife.29602.008

The following figure supplements are available for figure 3:

**Figure supplement 1.** All 188 single base-pair $M^2R$ quartets for *add* riboswitch.
DOI: https://doi.org/10.7554/eLife.29602.009

**Figure supplement 2.** All 82 double base-pair $M^2R$ quartets for *add* riboswitch.
DOI: https://doi.org/10.7554/eLife.29602.010

the denominators of *Equations (2) and (3)* – before then assessing whether the frequency rises or decreases in response to locking other helices through designed mutations. In addition, prior studies, including detailed NMR analysis, offered helix frequency data on the wild type *add* sequence that might validate or falsify our strategy of measuring rescue factors and using simulations to infer helix frequencies.

Overall, compensatory rescue experiments agreed well with prior models. In the absence of adenine, all prior models of the *add* riboswitch predicted that the RNA would show a diverse set of helices. Indeed, aptamer helices P1 (53%) and P2 (31%) were detectable, as they gave partial rescue in mutate-map-rescue experiments. For illustration, *Figure 3A* shows SHAPE profile quartets and

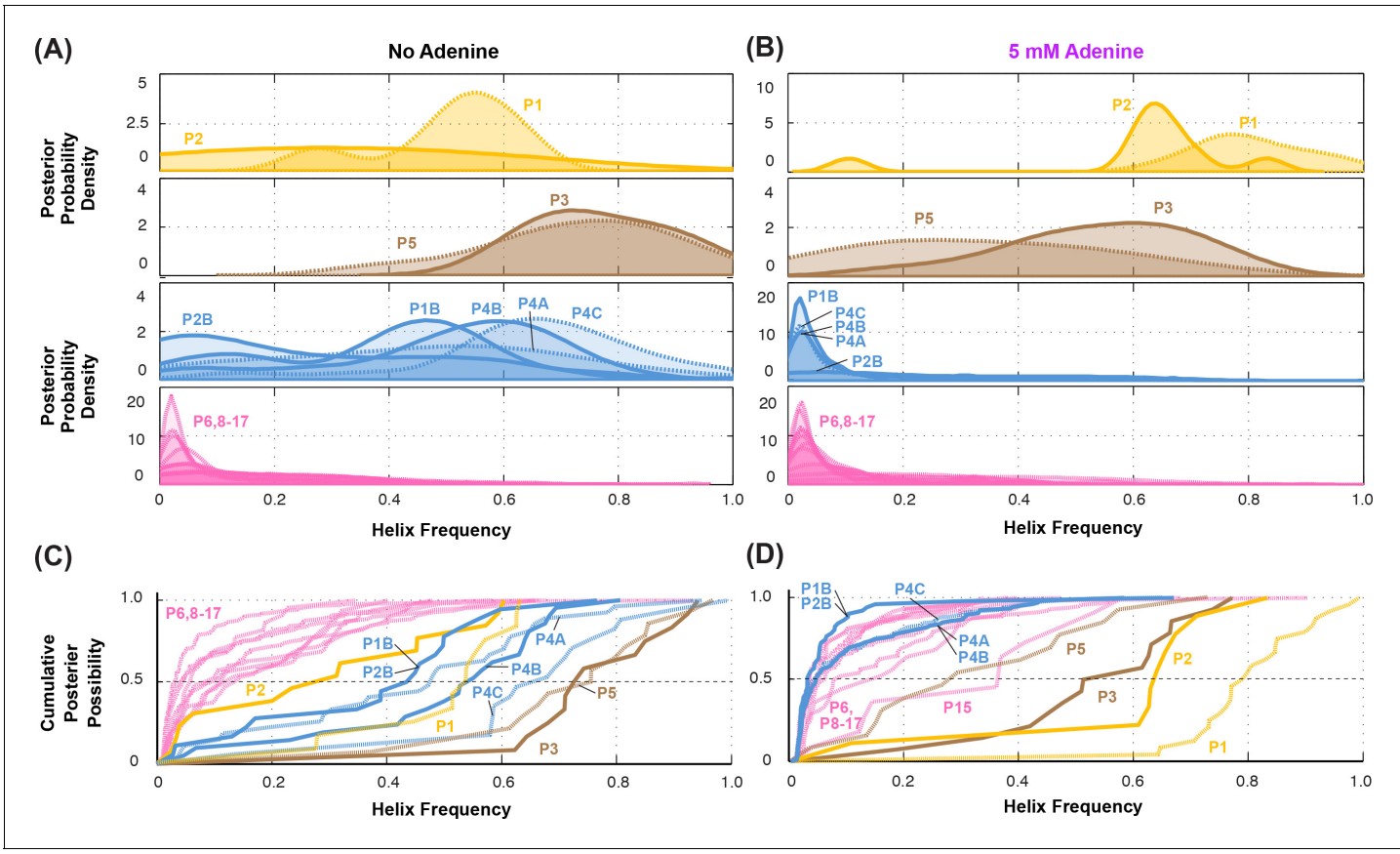

**Figure 4.** Summary of helix frequencies inferred for the *add* riboswitch. Posterior distributions over helix frequencies (A) without adenine ligand and (B) with 5 mM adenine ligand. Curves have been smoothed through kernel density estimation. Sub-panels separate helices expected to have high frequencies with adenine bound (gold); 'positive control' helices expected to be present irrespective of adenine (brown); helices expected to be present only in the absence of adenine (blue); additional helices that are not expected to have high frequencies (pink). (C-D) same distributions as in (A-B) but represented as cumulative distributions and overlaid together without smoothing.

DOI: https://doi.org/10.7554/eLife.29602.011

rescue factors for each base pair of P1; the inferred probability distributions over P1 helix frequencies is given as a gold curve underneath, and extends across helix frequencies ranging from 0.2 to 0.75. Helices P3 and P5 were expected to be present, and both were detected by compensatory rescue experiments, with median helix frequencies of 73% and 75%, respectively (brown curves, *Figure 4*; *Figure 3—figure supplement 1* gives SHAPE profiles). In addition, compensatory rescue experiments recovered evidence for helices P4A (48%), P4B (55%), P4C (68%), P1B (44%), and, at a lower level, P2B (17%), all proposed as part of *apoB* (*Figure 1*; left-hand structure in top panel, *Figure 3*; *Figure 3B* shows SHAPE profiles and associated partial rescue for each base pair of P4B). The posterior distributions of these helix frequencies, shown as blue curves in *Figure 4A and B*, were centered at intermediate frequencies, as expected. Other helices proposed to occur at low frequencies from M²-guided modeling (P6, P8 through P17) (*Cordero and Das, 2015*), gave no evidence for their presence (pink curves, *Figure 4*; *Figure 3—figure supplement 1*); those experiments thus served as negative controls for the M²R helix frequency estimation method. Overall, these measurements confirmed that the adenine-free state of the *add* riboswitch is quite heterogeneous, with evidence for intermediate frequencies for all helices predicted in prior studies and no further helices.

Turning to the adenine-bound state, all prior models predicted that the riboswitch would shift its helix frequencies to a better defined secondary structure with helices P1, P2, and P3 in the aptamer and P5 (*holo*, *Figure 1*; right-hand structure in top panel, *Figure 3*). M²R experiments testing each of these helices showed visually striking rescue in their SHAPE profiles as mutations disrupted adenine binding, and compensatory mutations restored the adenine-bound profiles (*Figure 3C* shows SHAPE profiles and rescue factors for each base pair of P1). Indeed, in the presence of 5 mM adenine, SHAPE measurements for single and compensatory mutations targeting these helices and rescue factor analysis gave helix frequencies of 79%, 64%, 56% and 29%, respectively (*Figure 3C*; and gold and brown curves in *Figure 4C and D*). Measuring rescue factors for other helices, including P1B, P2B, P4A, P4B, and P4C predicted for the adenine-free state and other helices proposed to occur at low frequencies from M²-guided modeling (P6, P8 through P17) (*Cordero and Das, 2015*), gave no evidence for their presence when adenine was bound (example SHAPE profile quartets for P4B in *Figure 3D*; all quartets in *Figure 3—figure supplement 1*; and blue and pink curves, *Figure 4C and D*). In distributions of posterior probabilities, there was a clear separation of these low-frequency helices from the high-frequency helices (*Figure 4C and D*; and *Table 1*). Helices P6, P8 through P17 (*Figure 3—figure supplement 2*) again served as negative controls and gave low helix frequencies by compensatory rescue, as expected (pink distributions in *Figure 3C and D*; *Table 1*). In addition to these measurements based on mutations of each base pair of a helix, we used a double-base-pair mutation scheme for M²R, which induced stronger perturbations and rescuing effects, and supported the same pairings as above (*Figure 3—figure supplement 2*). Overall, M²R data and helix frequency analysis for the *apo* and *holo* ensembles detected the helices expected from prior studies and provided baseline helix frequency values to infer the correlation or anticorrelation of each pair of these helices.

## Four-dimensional Lock-Mutate-Map-Rescue enables measurement of helix-helix correlations

The measurements above showed that locking the aptamer secondary structure through adenine binding reduced the frequencies of helices characteristic of the closing of the gene expression platform (compare upward shift of gold curves – and downward shift of blue curves – between *Figure 4A and C*). Discrimination between MWC and non-MWC models involves determining whether this same allosteric communication between helices occurs even in the absence of adenine ligand. We therefore sought to measure whether locking helices characteristic of the aptamer secondary structure through mutation, rather than adenine binding, also reduced the frequencies of helices characteristic of the closing of the gene expression platform, and vice versa.

To identify appropriate locking mutants, we noted that a number of double-base-pair compensatory mutants for each of P1, P2, P1A, P4A, and P4B (Figure 3—figure supplement 2) exhibited SHAPE profiles that differed from WT RNA but agreed with each other, suggesting that they had independently locked the same state (*Figure 5*; data for all tested mutants given in *Figure 5—figure supplement 1*). For example, numerous P1- or P2-locking mutants exhibited increased reactivity in the distal gene expression platform region (nts 105–122) relative to the adenine-free WT RNA, at a level comparable to the adenine-bound WT RNA. This observation already suggested that mutation-

**Table 1.** Helix frequency estimates for single base-pair M²R, double base-pair M²R, LM²R experiments.

Median helix frequencies from Rfam simulations corresponding to each experimentally observed rescue factor are reported. The experimentally observed rescue factor for each helix was averaged across all tested base pairs. Full posterior distributions are presented in main text *Figure 4* and *Figure 7—figure supplement 1*.

| Wild type* | | | In locked contexts† | | |
|---|---|---|---|---|---|
| [adenine] | 0 mM | 5 mM | | | 0 mM |
| P1 | 53% | 79% | Lock P1 | P2 | 42% |
| P2 | 31% | 64% | | P4B | 4% |
| P3 | 73% | 56% | Lock P2 | P1 | 76% |
| P5 | 75% | 29% | | P4A | 4% |
| P1B | 44% | 3% | | P4B | 4% |
| P2B | 17% | 17% | Lock P4A | P1B | 9% |
| P4A | 48% | 4% | | P2B | 6% |
| P4B | 55% | 4% | | P2 | 6% |
| P4C | 68% | 4% | | P4B | 55% |
| P6 | 6% | 5% | Lock P1B | P2B | 53% |
| P8 | 9% | 4% | | P4A | 55% |
| P9 | 12% | 3% | | P4B | 80% |
| P10 | 3% | 3% | | P3 | 82% |
| P11 | 12% | 16% | Lock P4B | P1B | 8% |
| P12 | 10% | 6% | | P2B | 4% |
| P13 | 9% | 4% | | P1 | 5% |
| P14 | 4% | 4% | | P2 | 6% |
| P15 | 12% | 36% | | P4A | 47% |
| P16 | 14% | 8% | Mut P2 | P1 | 38% |
| P17 | 4% | 4% | | P4A | 8% |
| | | | | P4B | 13% |

*Median helix frequencies inferred from mutate-map-rescue, compensatory rescue read out by chemical mapping across the transcript.

†Lock-mutate-map-rescue, mutate-map-rescue carried out in the context of mutations that 'lock' the specified helices.

DOI: https://doi.org/10.7554/eLife.29602.015

based locking of P1 or P2 opens the gene expression platform, favoring the MWC model over the non-MWC model, although it depends on an assumption that increase in SHAPE reactivity correlates with reduced base pairing, which does not always hold (*Cordero et al., 2012a*; *Deng et al., 2016*). We also observed similarities in the gene expression platform between these adenine-free P1 and P2-stabilizing mutants and adenine-bound WT RNA in profiles acquired with chemical modifiers besides SHAPE reagents (dimethyl sulfate, glyoxal, ribonuclease V1, terbium(III), and ultraviolet irradiation at 302 nm; *Figure 5—figure supplement 2*). For other helices P1B, P4A, and P4B, we were able to find double-base-pair mutants that gave similar SHAPE profiles to each other, supporting the use of those mutants as helix-locked variants (*Figure 5*). Interestingly, for each of these three target helices, there was one double-base-pair mutant with spikes in SHAPE reactivity at positions 78 and 85, which also appear in the wild type RNA but not the P1 or P2-lock mutants (red arrows, *Figure 5*). This observation suggested that these double-base-pair mutants for P1B, P4A, and P4B stabilized the same state; we therefore chose these variants as the lock mutants. Further evidence that these mutants were appropriately locked came from excellent two-state fits that recovered the adenine-free wild type chemical mapping profiles across five modifiers from the profile for any of these

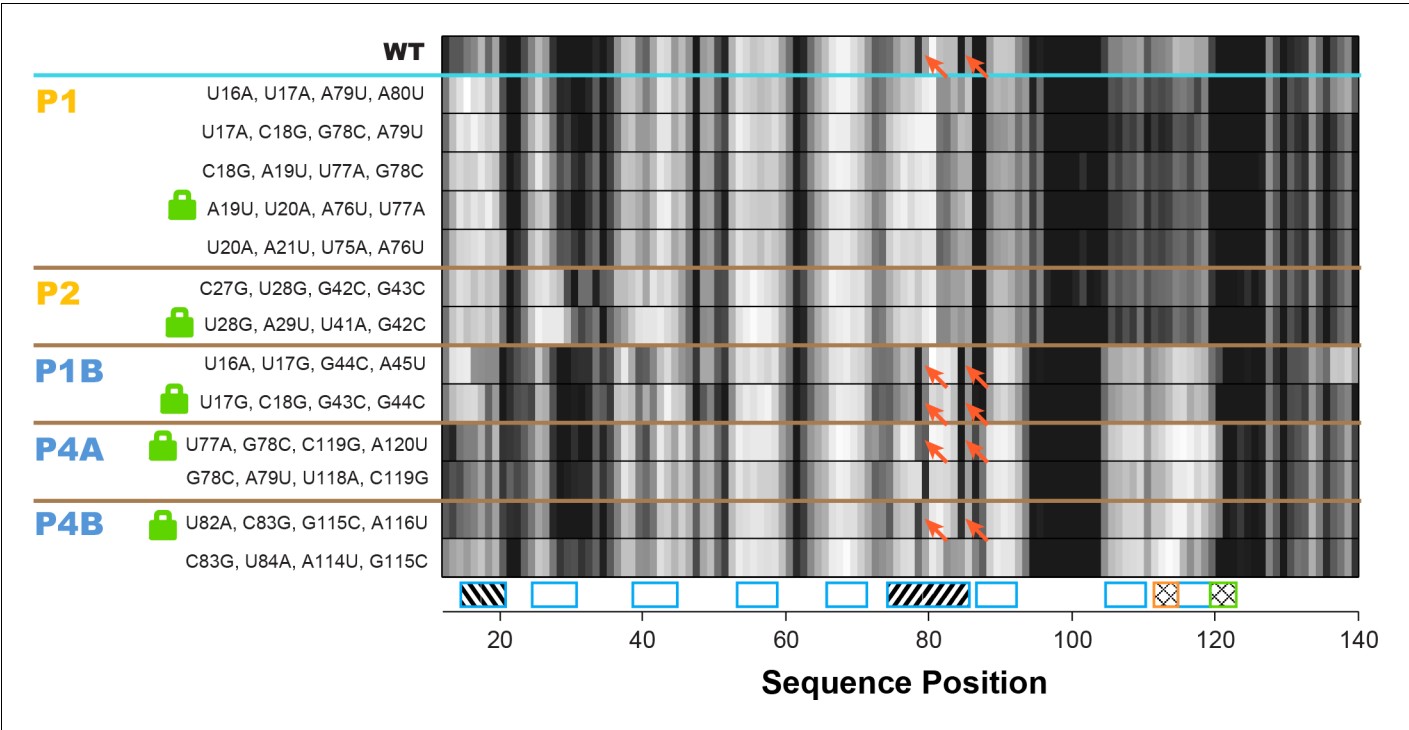

**Figure 5.** Double-base-pair mutations to lock each helix of the adenine riboswitch. One-dimensional SHAPE reactivity of candidate constructs for locking each *add* riboswitch helix. In each candidate, two consecutive base pairs of the helix were switched to alternative Watson-Crick pairs (*Figure 3—supplement 2*). For each helix targeted, at least two sets of mutations gave SHAPE reactivities distinct from wild type but similar to each other. Red arrows mark residues in switch region that are reactive in wild type and strongly reactive in lock mutants chosen for P1B, P4A, and P4B.
DOI: https://doi.org/10.7554/eLife.29602.012

The following figure supplements are available for figure 5:

**Figure supplement 1.** All double-base-pair mutations tested as possible locking mutations.
DOI: https://doi.org/10.7554/eLife.29602.013

**Figure supplement 2.** Multi-probe chemical mapping of stabilizers and linear fitting of P1 and P4A.
DOI: https://doi.org/10.7554/eLife.29602.014

P1B- P4A- or P4B- lock mutants with the profiles for any of the P1 or P2-locking mutants (*Figure 5—figure supplement 2*). (P2B and P4C were short two-base-pair helices; since only one candidate double-base-pair lock mutation could be tested and did not allow such cross-checks, we chose to not go forward with P2B and P4C locking experiments.)

Compensatory mutation experiments in locked mutation backgrounds allowed us to confidently determine whether MWC-predicted anticorrelations exist in the adenine-free structural ensemble of the riboswitch. *Figure 6A* shows M$^2$R data for the P1 helix in the context of P4B-stabilizing mutations (lock-P4B; U82A, C83G, G115C, A116U). In contrast to the observation of partial rescue of P1 in the wild type RNA (*Figure 3A*), these data show no rescue for P1 in the lock-P4B mutational background. The inferred P4B helix frequency drops from 53%% to 5%; the shift in posterior probability distribution to lower helix frequency values is clear (faded gold to gold curves; *Figure 6B*). Stated simply, locking P4B strongly reduces P1. These results support the anticorrelation between P1 and P4B expected from the MWC model, as opposed to the independence of the helices expected from the non-MWC model.

To statistically evaluate the MWC vs. non-MWC model, we computed the correlation value $g(P1, P4B)$ through *Equation (2)*, taking into account the full uncertainties in helix frequencies shown in *Figure 6B*. The resulting posterior distribution for $g(P1, P4B)$ (*Figure 7A*) is highly skewed away from 1 (the expectation of the non-MWC model) to values close to zero (expectation of the MWC model). Statistical significance in biological studies is often summarized through a *p*-value for a null hypothesis. In our Bayesian framework, an analogous value is the total posterior support for non-

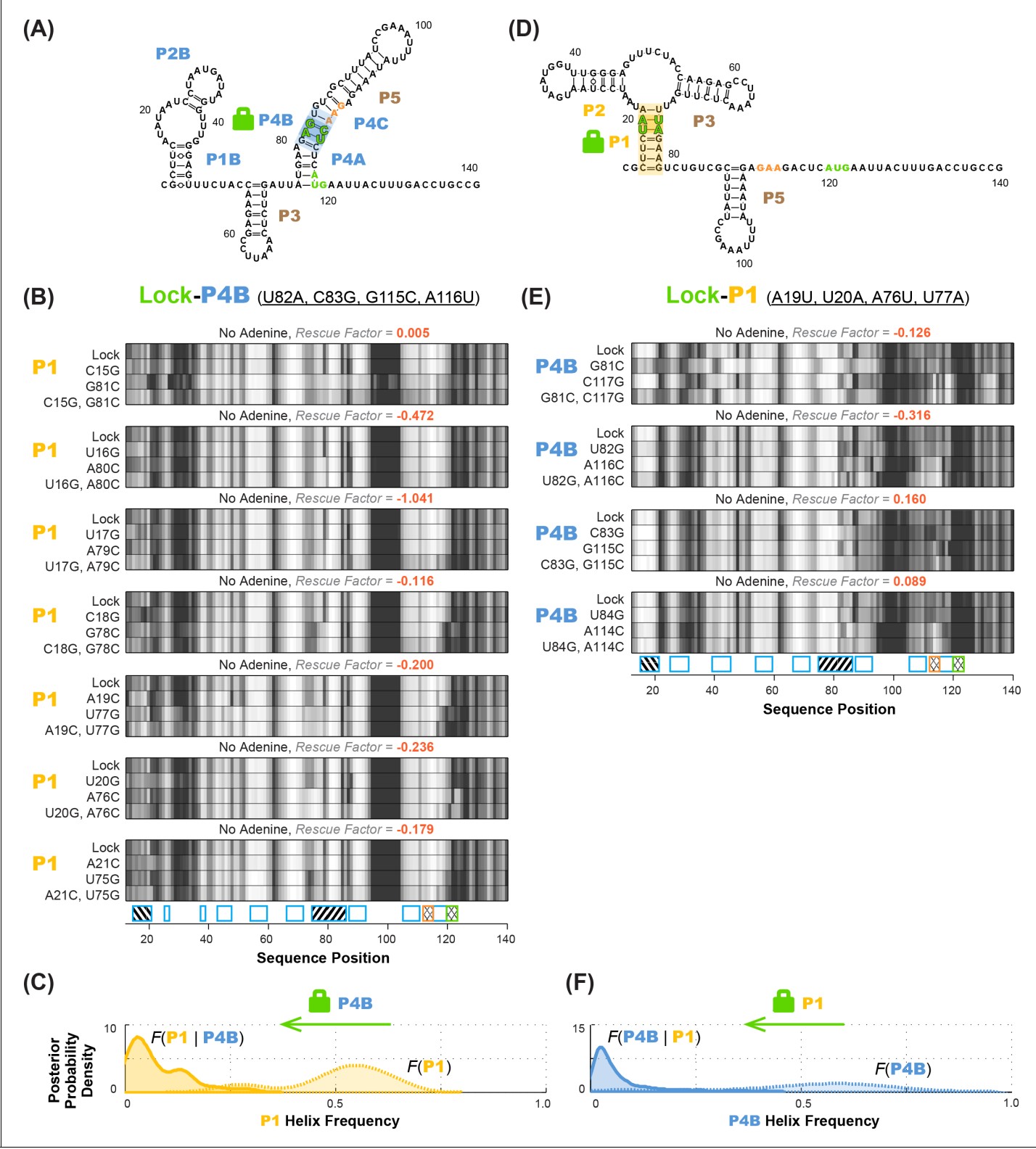

**Figure 6.** Anticorrelation between P1 and P4B helices in ligand-free *add* riboswitch structural ensemble, inferred through lock-mutate-map-rescue (M$^2$R). (**A–C**) M$^2$R quartets probing P1 in the context of lock-P4B mutations show no rescue. (**D–F**) M$^2$R quartets probing P4B in the context of lock-P1 mutations show no rescue. Panels (**A,D**) show possible secondary structures for helix lock mutants. Panels (**B,E**) give M$^2$R quartets for each base pair of the probed helix in the context of the mutations locking the other helix. Panels (**C,F**) give posterior probability distributions over helix frequency, estimated from the experimental M$^2$R rescue factors in locked background and wild type background (solid and dotted curves, respectively).

*Figure 6 continued on next page*

*Figure 6 continued*

DOI: https://doi.org/10.7554/eLife.29602.016

The following figure supplements are available for figure 6:

**Figure supplement 1.** All 82 single base-pair LM²R quartets for *add* riboswitch.

DOI: https://doi.org/10.7554/eLife.29602.017

**Figure supplement 2.** All 15 single base-pair LM²R quartets for *add* riboswitch with the MutP2 construct.

DOI: https://doi.org/10.7554/eLife.29602.018

MWC models, i.e. the integral of the posterior probability distribution in *Figure 7A* for $g(\mathrm{P1, P4B})>1$. This posterior support is small, with a value of $3.1 \times 10^{-3}$, disfavoring the non-MWC model and supporting the MWC model.

As an independent test of the MWC vs. non-MWC models, we carried out the 'flipped' P1-P4B lock-mutate-map-rescue (LM²R) experiments, in which we locked P1 through mutations (lock-P1; A19U, U20A, A76U, U77A) and assessed the helix frequency of P4B. Again, in contrast to the observation of partial rescue of P4B in the wild type RNA (*Figure 3B*), the data in the lock-P1 context show no rescue for P4B. Upon locking P1, the P4B helix frequency drops, from 55% to 4%, and there is a striking shift in the posterior probability distribution to lower helix frequencies (faded blue to blue curves; *Figure 6D*). The posterior support for the non-MWC model [$g(\mathrm{P1, P4B})>1$] from this 'flipped' lock-P1 experiment is somewhat higher than the lock-P4B experiment above, but still small (0.039; full posterior distribution in *Figure 7B*). If the data for the two experiments are combined, the posterior support for the non-MWC model becomes quite small, $4 \times 10^{-5}$. Another way to summarize the results is to estimate whether the correlation value $g(\mathrm{P1, P4B})$ for P1 and P4B is not just

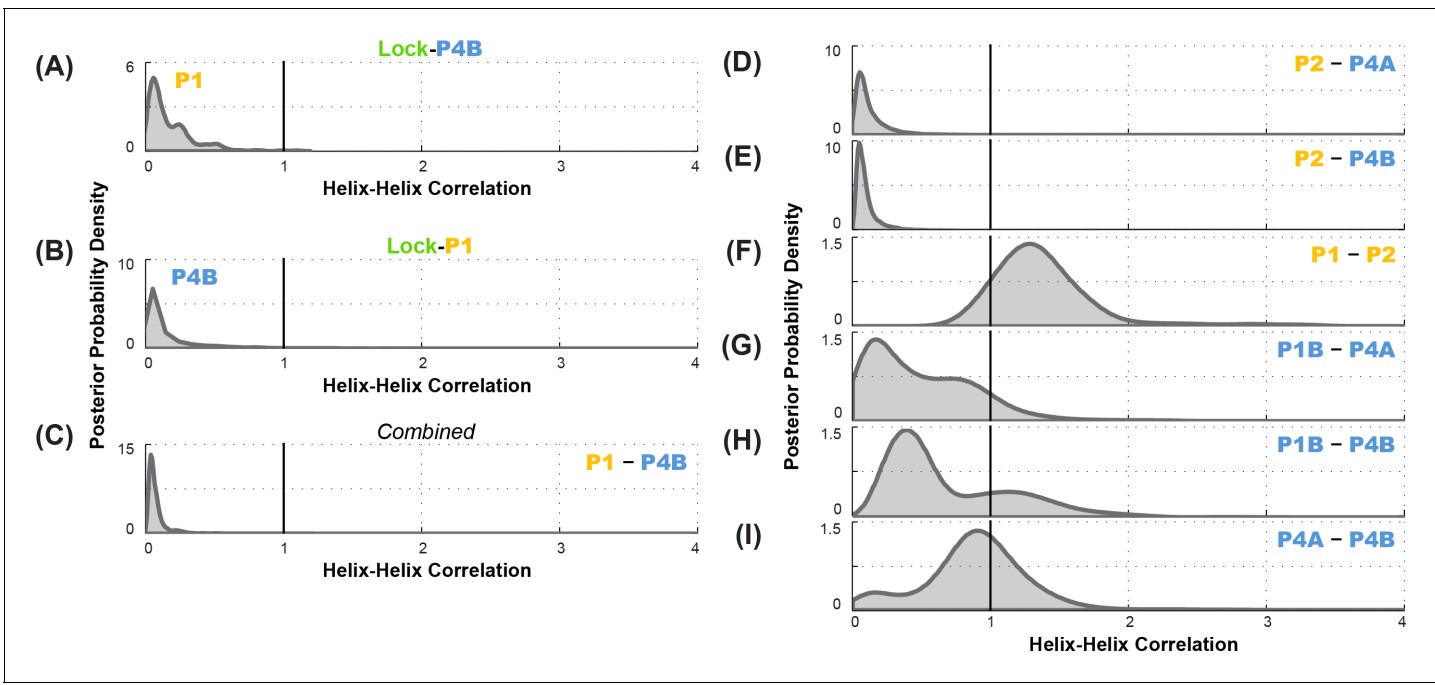

**Figure 7.** Correlation analysis for functionally important helix-helix pairs in the ligand-free *add* riboswitch structural ensemble. Posterior probability distributions, smoothed through kernel density estimation, for correlation of (A-C) P1 and P4B, based on LM²R experiments (A) locking P4B and probing P1, (B) locking P1 and probing P4B, and (C) combination of experiments of (A) and (B); and (D-I) other pairs of helices, based on combination of experiments locking each helix and probing the other and vice versa. Helix pairs are (D) P2 and P4A, (E) P2 and P4B, (F) P1 and P2; (G) P1B and P4A; (H) P1B and P4B; and (I) P4A and P4B.

DOI: https://doi.org/10.7554/eLife.29602.019

The following figure supplement is available for figure 7:

**Figure supplement 1.** Posterior probability distributions over helix frequencies and helix-helix correlations for individual LM²R experiments.

DOI: https://doi.org/10.7554/eLife.29602.020

less than one but much less than 1 – a strong anticorrelation. Indeed, the full posterior distribution in *Figure 7C* is highly skewed towards zero. The median for this correlation value is 0.052 (*Table 2*), and with 95% posterior support, $g(P1, P4B)$ is less than 0.2. The data agree well with strong anticorrelation predicted by the MWC model but not the non-MWC model.

Further data support the MWC-predicted anticorrelation of aptamer secondary structure helices and the closure of the gene expression platform. LM²R experiments testing correlation of P1 and P4A were not possible due to overlap of nucleotides between the helices; locking one helix precludes compensatory mutagenesis of base pairs in the other helix. However, we could carry out complete LM²R experiments probing the correlation between P2 and P4B and the correlation between P2 and P4A. Full data are given in *Figure 6—figure supplement 1*, and final posterior distributions for $g(P2, P4A)$ and $g(P2, P4B)$ are shown in *Figure 7D* and *Figure 7E*. Both sets of experiments again showed strong skewing of these two correlation values towards zero (anticorrelation; median correlation values of 0.069 and 0.089, respectively; see *Table 2*). These results again strongly disfavor the non-MWC model (posterior support: $4.1 \times 10^{-3}$ and $1.5 \times 10^{-3}$, respectively) and supporting the MWC model. We emphasize that these two sets of experiments are independent of each other and of the $g(P1, P4B)$ experiments above.

To gain further insights into the *apo* ensemble, we also carried out LM²R experiments for pairs of helices that were predicted to co-occur (correlate, rather than anticorrelate) in prior models of the *apo* ensemble (*Figure 1*). On one hand, LM²R experiments gave evidence of weak correlation between the two helices characteristic of the aptamer secondary structure, P1 and P2: $g(P1, P2)$ is inferred to be somewhat greater than 1 (*Figure 7F*). On the other hand, we saw no evidence of correlation for pairs of helices expected to co-occur in the riboswitch when the aptamer secondary structure is not formed (*apoB*, *Figure 7G*). LM²R experiments gave posterior distributions for $g(P1B, P4A)$, $g(P1B, P4B)$, and $g(P4A, P4B)$ centered around 1 (*Figure 7G–I*), indicating no correlation between the occurrence of these helices in the ligand-free ensemble. While the uncertainties in these LM²R measurements allow correlation values of these helices to be slightly greater than 1, the data are most consistent with a model in which helices P4A, P1B, P4B each appear with significant frequency in the *apo* ensemble, but do not necessarily co-occur, resulting a heterogeneous structural ensemble. Therefore, a single secondary structure such as *apoB* may not be an appropriate description of the structural ensemble. Further evidence for structural heterogeneity came from the incomplete match in SHAPE profiles for mutants that lock different helices of the putative *apoB* state. For example, lock-P4A and lock-P1B showed reproducible differences in SHAPE reactivities at the 5′ end of the RNA (nts 13–20) that were larger than, e.g., differences between lock-P1 and lock-P2 (*Figure 5*), and we did not discover any alternative locking mutations that brought them into agreement.

**Table 2.** Helix-helix correlation estimates from LM²R experiments.
Median values are reported. Full posterior distributions are presented in main text *Figure 6* and *Figure 7—figure supplement 1*.

| Helix-helix | Correlation value |
| --- | --- |
| P1-P4B | 0.052 |
| P2-P4B | 0.069 |
| P2-P4A | 0.089 |
| P1-P2 | 1.315 |
| P4A-P4B | 0.902 |
| P1B-P4A | 0.44 |
| P1B-P4B | 0.534 |

DOI: https://doi.org/10.7554/eLife.29602.025

## Discussion

### Four-dimensional chemical mapping for complex RNA ensembles

Complex secondary structure ensembles underlie many, and perhaps most, RNA regulatory elements. Understanding how a panoply of structures underlies the allosteric mechanism of these elements requires assessing not just the frequencies of different helices in the RNA structural ensemble but how the presence of one helix enhances or suppresses the frequency of others. This study has introduced a 'four-dimensional' expansion of chemical mapping that offers an experimental route to this information. The strategy is to 'lock' one helix into place through mutations (dimension 1) and to then introduce mutations elsewhere (dimension 2). Perturbation of the chemical mapping profile at other nucleotides (dimension 3) provides evidence of involvement by the mutated nucleotide in RNA structure, perhaps a second helix. If compensatory mutation of a candidate partner in the second helix (dimension 4) then gives no, partial, or complete rescue of the mapping profile, the frequency of the second helix can be inferred to be low, medium, or high, with uncertainties estimated from simulation. Importantly, quantitative inferences of helix-helix correlations from this lock-mutate-map-rescue (LM$^2$R) workflow do not require interpretation of how SHAPE reactivity corresponds to Watson-Crick pairing probability, which can lead to erroneous conclusions (*Tian et al., 2014*). In that sense, the method is analogous to three-dimensional and four-dimensional pulse sequences that enable structural conclusions to be derived from NMR. Inferences from multidimensional NMR are based on the changes of spectroscopic signals (chemical shifts, magnetization transfer rates) rather than on absolute values of those signals, whose relationships to features of macromolecule structure are not yet understood in quantitative detail (*Wüthrich, 2003*).

The LM$^2$R method currently has limitations regarding the length of RNAs that can be probed. The current throughput of PCR assembly and capillary electrophoretic sequencing enables single-nucleotide-resolution mapping of approximately 192 constructs per experiment at the signal-to-noise needed for statistically sound conclusions. With this throughput, in vitro dissection of the co-occurrence or anticorrelation of helices underlying a ∼ 120 nucleotide RNA domain was feasible over a 1–2 month period. However, for RNAs much longer than 200 nucleotides, the expense and time of preparing mutations would likely become prohibitive.

The LM$^2$R method also currently has limitations regarding which kinds of functional elements can be interrogated for correlation or anticorrelation. Our study has focused on helix elements, which can be selectively probed and selectively locked through mutation of helix base pairs. However, other functional elements in RNA folding and gene regulation include single-stranded regions and non-Watson-Crick pairings that underlie tertiary structure. For example, in the *add* riboswitch the functional elements in gene expression are the single-stranded AUG and Shine-Dalgarno ribosome binding site. We used the helices in the P4 domain to allow locking and probing (by compensatory rescue) of this region. A more general approach to lock and probe such single-stranded elements might be to sequester them with designed oligonucleotide probes and to assess dissociation constants for such probes, respectively. For tertiary contacts or non-Watson-Crick paired junctions, it is less clear how to generally lock the elements or to carry out compensatory mutagenesis to probe the elements, in which case, LM$^2$R may need to continue relying on Watson-Crick helices near these elements as proxies.

As a final caveat, the current LM$^2$R method relies on the RNA system being at equilibrium so that the correlation *Equations (1)-(3)* are valid and also so that extensive computer simulations can be carried out that allow conversion of observed rescue factors to helix frequencies. It will be important to evaluate whether LM$^2$R, or some time-dependent extension of the method, could be applied to RNAs that change their structures in non-equilibrium scenarios involving co-transcriptional RNA folding, recruitment of energy-dissipating machines like the ribosome or RNA polymerase, or other behaviors that depend on intrinsically kinetic mechanisms (*Yanofsky, 2000*; *Watters et al., 2016*).

### *Resolving the core mechanism of the* add *riboswitch*

This work demonstrates application of four-dimensional chemical mapping to resolve a fundamental biochemical question. Applying the methodology to the *add* riboswitch from *V. vulnificus* supports a Monod-Wyman-Changeux (MWC, or population shift, or conformational selection) model of allostery rather than a non-MWC revision proposed after detailed NMR experiments (*Reining et al., 2013*;

*Fürtig et al., 2015a*). The two prior models differed in one core aspect: whether structuring of the molecule's aptamer region is allosterically communicated to expose a ribosome binding site and turn on *add* mRNA gene expression, even in the absence of adenine (*Figure 1C and D*). In LM²R experiments locking P1 and P2, which are specific to the aptameric secondary structure, the rescue data showed depletion of P4 helices that would sequester the ribosome binding site. In other words, the *add* gene expression platform is ON when the sequence-separate aptamer region is folded even in the absence of the adenine ligand (*Figure 1A and C*; *Figure 8, apoA*). Conversely, LM²R experiments locking P1B, P4A, or P4B turn the riboswitch OFF and give evidence against the formation of P1 and P2 helices.

These data favor anti-correlation between aptamer helix formation and the structures sequestering the ribosome binding site even without adenine binding, as predicted in the MWC conformational selection model, and disfavor a non-MWC revision that omits this coupling (compare *Figure 1A* and *Figure 1B*; or *Figure 1C* and *Figure 1D*). Stated differently, allosteric communication of the aptameric region and the gene expression platform is an intrinsic property of the folding

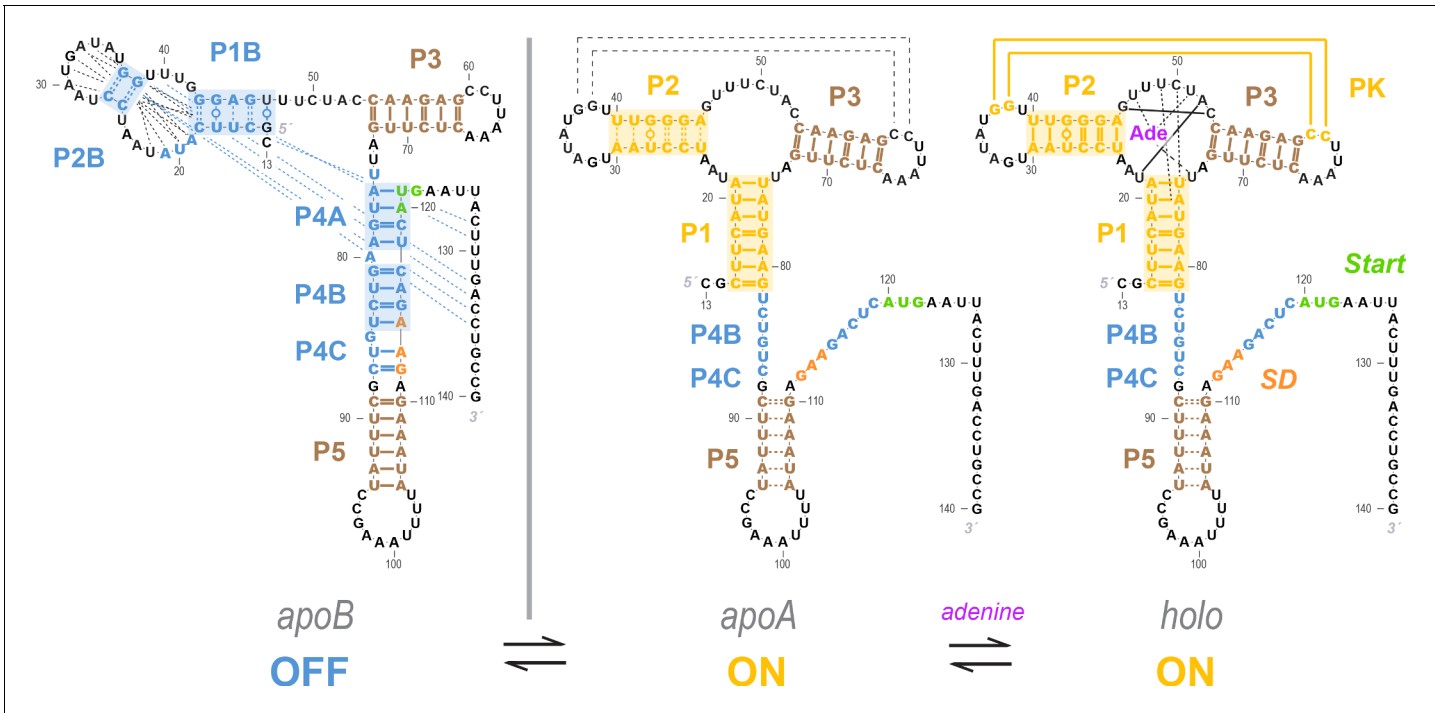

**Figure 8.** Current model for the *add* riboswitch structural ensemble. The proposed model favors a Monod-Wyman-Changeux (population shift, conformational selection) model of allostery. The ligand-free *apoB* state (left) sequesters the Shine-Dalgarno sequence and AUG codon in the P4 domain, precluding gene expression. Our measurements show that sampling of the aptamer secondary structure is strongly anticorrelated with formation of the P4 domain; it is therefore valid to represent it as a separate state (*apoA*, middle). This *apoA* state is structurally similar to the ligand-bound *holo*, including not just the aptamer region but also the gene expression platform, as predicted by an MWC framework (right). Our measurements and other studies are consistent with sampling of additional alternative structures in the *apoB* state (dashed blue lines), a partially formed P5 in *apoA*, and additional stabilization of the aptamer through coupled tertiary contacts upon adenine binding (dashed black lines in *apoA*, solid yellow lines in *holo*).

DOI: https://doi.org/10.7554/eLife.29602.021

The following figure supplements are available for figure 8:

**Figure supplement 1.** Simulation of temperature dependence on switching efficiency.
DOI: https://doi.org/10.7554/eLife.29602.022

**Figure supplement 2.** Models of additional structures that the *add* riboswitch can form, based on mutate-and-map on selected single mutant backgrounds.
DOI: https://doi.org/10.7554/eLife.29602.023

**Figure supplement 3.** Higher-order mutate-and-map on selected single mutant backgrounds.
DOI: https://doi.org/10.7554/eLife.29602.024

landscape of the *add* riboswitch sequence that is established without adenine. Despite its similarity to MWC allosteric mechanisms for proteins, this model for the *add* riboswitch shows much more dramatic changes in structure than is typically seen in protein allostery, where conformational shifts typically preserve secondary structure (*Changeux and Edelstein, 2005*; *Eaton et al., 1999*).

## Comparison to prior data and proposals

*Figure 8* summarizes our current model of the *add* riboswitch. Our results indicate a strong anticorrelation of helices P4A and P4B with the helices P1 and P2, and so we continue to show the *apo* ensemble as split between *apoB* and *apoA* of the MWC model (*Figure 1*), but now highlight alternative or partial pairings (dashed lines) to better convey the likely heterogeneity of these states. This model is consistent with all data collected in this study as well as all measurements on the *V. vulnificus add* riboswitch published to date. First, a number of prior studies applied techniques that do not directly read out base pairing but are sensitive to ligand binding and can quantify energetics and kinetics. Functional measurements in vitro and in vivo, use of fluorescent reporters, and single-molecule force experiments all have suggested that the riboswitch is rather 'leaky', forming the aptamer secondary structure at ~40% frequency at 20–30°C, even without adenine (*Lemay et al., 2011*; *Mandal and Breaker, 2004*; *Batey, 2012*; *Cordero and Das, 2015*; *Stoddard et al., 2013*; *Rieder et al., 2007*; *Neupane et al., 2011*; *Lemay et al., 2006*; *Greenleaf et al., 2008*; *Wickiser et al., 2005*; *Lemay and Lafontaine, 2007*; *Reining et al., 2013*). Second, X-ray crystallographic studies have been limited to constructs with just the aptameric region (*Stagno et al., 2017*; *Stoddard et al., 2013b*; *Zhang and Ferré-D'Amaré, 2014*; *Gilbert et al., 2009*; *Edwards and Batey, 2009*; *Serganov et al., 2004*; *Daldrop et al., 2011*; *Delfosse et al., 2010*; *Gilbert et al., 2007*); these structures all show P1, P2, and P3 helices, in accord with *Figure 8*. Third, the detailed NMR measurements that suggested the alternative *apoA* (*Figure 1B*) were also consistent with the *apoA* base pairings in the MWC model (*Figure 1A*): nuclear Overhauser effect (NOE) spectroscopy unambiguously established that pairings in both P1 and P4A occur in the absence of adenine, but could not infer whether these helices might co-occur or be mutually exclusive; our LM$^2$R data strongly support the latter scenario (*Figures 6* and *7*). Fourth, analogous to our locking approach, the NMR study used a variant MutP2 (A29C, A30G, U40C, U41G) to stabilize the aptamer secondary structure. While our chemical mapping measurements suggest that this mutant only partially stabilizes P2 (*Figure 5—figure supplement 1*), both the NMR study and our LM$^2$R measurements detect suppression of P4A in this background (*Figure 6—figure supplement 2*), supporting the anticorrelation between P2 and P4A in *Figure 8*. Fifth, our SHAPE and multi-probe chemical mapping data suggest partial opening of P5 in *apoA* and *holo* (brown curves, *Figure 4B*), and our LM$^2$R analysis suggests only partial formation of P2B and P1B when P4A is locked (*Figure 6—figure supplement 1* and *Figure 7—figure supplement 1D*). These observations are again consistent with NMR studies which could unambiguously detect P5 and estimate relative frequencies of, e.g., P2 vs. P2B, but could not establish these helices' absolute frequencies. Sixth, newer single molecule FRET measurements detect transient formation of the L2/L3 tertiary contact prior to adenine binding (dashed lines in *apoA* of *Figure 6*) and stabilization upon adenine binding to *holo* (*Warhaut et al., 2017*). These data are consistent with, and indeed were interpreted within, an MWC framework in which formation of the aptameric secondary structure occurs in concert with opening of the gene expression platform, even without adenine ligand present (*Warhaut et al., 2017*). Seventh, the prior detailed NMR analysis led to a compelling model of riboswitch temperature robustness that involves compensating the improving affinity of the aptamer for adenine at colder temperatures by a pre-equilibrium that favored *apoB* over *apoA* at those temperatures. Although the precise predictions of the switching efficiencies of this model are affected by whether *apoA* involves opening the gene expression platform, approximately the same temperature compensation occurs in the MWC conformational selection model (see *Figure 8—figure supplement 1*).

There is now concordance across numerous measurements for an MWC model that posits strong secondary structure similarities between an *apoA* state that transiently forms the aptamer secondary structure without adenine ligand and the ligand-bound *holo* state. We also propose herein that further heterogeneity is likely to be present in members of the *apo* ensemble that do not present the aptamer secondary structure, grouped into *apoB*. LM$^2$R measurements indicate weak or no correlation of the tested signature helices of *apoB* (P1B, P4A, and P4B; *Figure 7*). Furthermore, numerous alternative secondary structures are possible for this state (see, e.g., *Figure 1—figure supplement*

*3*). The presence of these myriad alternative helices, each at low population, would explain their detection difficulty with this and prior bulk equilibrium techniques but their appearance in single-molecule measurements (*Neupane et al., 2011*). Supporting this picture, higher-order $M^2$ analysis in single mutant backgrounds isolate and confirm numerous alternative secondary structures for the *add* riboswitch (*Figure 8—figure supplement 2* and *Figure 8—figure supplement 3*). We note that there are no functional reasons for the *apoB* ensemble to maintain a single secondary structure – it simply has to disallow adenine binding while keeping the gene expression platform closed in P4. Without selection for a pure single structure, we therefore suggest that *apoB* has remained structurally heterogeneous, and the exact populations of its helix pairings outside the P4 domain may shift in different solution conditions and flanking sequences while still being compatible with riboswitch function. We further speculate that 'non-functional' states of other riboswitches and RNA gene regulatory modules may have highly heterogeneous structures and, indeed, this feature might explain why those states have been refractory to conventional structural biology approaches developed primarily to dissect protein structure/function. This picture also implies that antibiotics targeting any specific *apo* structure of the *add* and other riboswitches are unlikely to succeed, as single mutations that disrupt the targeted *apo* structure while retaining other members of the *apo* ensemble with the P4 domain could evolve quickly and offer resistance.

### General applicability of 4D RNA chemical mapping

The next challenge for understanding riboswitches and other *cis*-regulatory RNA elements is to test how the structural ensembles defined through in vitro studies are retained or altered by co-transcriptional effects, protein binding, helicases, crowding, noise due to low numbers of molecules, and other complexities of these molecules' native biological environments (*Tian and Das, 2016*; *Leamy et al., 2016*). Amongst available biochemical and biophysical techniques, chemical mapping methods read out by sequencing have strong promise in delivering single-nucleotide-resolution structural information in such environments. As techniques improve to edit genomes across organisms and to amplify and measure RNA chemical mapping signals in cells and tissues, the lock-mutate-map-rescue approach developed here offers the possibility of dissecting complex structural ensembles for these shape-shifting molecules in situ.

## Materials and methods

### RNA synthesis and construct design

Double-stranded DNA templates were prepared by PCR assembly of DNA oligomers with maximum length of 60 nt ordered from IDT (Integrated DNA Technologies). DNA templates contain a 20-nt T7 RNA polymerase promoter sequence (TTCTAATACGACTCACTATA) on the 5′ end and a 20-nt Tail2 sequence (AAAGAAACAACAACAACAAC) on the 3′ end. The sequence of interest is flanked at each end by a hairpin with single-stranded buffering (*Kladwang et al., 2014*). The full sequence probed was:

GGACACGACUCGAGUAGAGUCGAAUGCGCUUCAUAUAAUCCUAAUGAUAUGGUUUGGGAG UUUCUACCAAGAGCCUUAAACUCUUGAUUAUGAAGUCUGUCGCUUUAUCCGAAAUUUUA UAAAGAGAAGACUCAUGAAUUACUUUGACCUGCCGACCGGAGUCGAGUAGACUCCAACAAAA-GAAACAACAACAACAAC.

Hairpin sequences are in italics; *add* riboswitch sequence is underlined. The primer assembly scheme and plate orders for all constructs were automatically designed by Primerize-2D (*Tian and Das, 2017a*).

PCR reactions, including 100 pmol of terminal primers and 1 pmol of internal primers were carried out as previously described. PCR products were purified using Ampure XP magnetic beads (Agencourt) on a 96-well microplate format following manufacturer's instructions. DNA concentrations were measured on a Nanodrop 1000 spectrophotometer (Thermo Scientific). In vitro transcription reactions were described previously, followed by similar purification (using Ampure XP beads with externally added 10% PEG-8000) and quantification steps.

## Chemical modification

$M^2$, $M^2R$ and $LM^2R$ chemical mapping were carried out in 96-well format as described previously (*Tian et al., 2014*; *Kladwang et al., 2011*). Prior to chemical modification, 1.2 pmol of RNA was heated to 90°C for 2 min and cooled on ice for 2 min to remove secondary structure heterogeneity, then folded for 20 min at 37°C in 15 μL of 10 mM $MgCl_2$, 50 mM Na-HEPES pH 8.0 (with or without 5 mM adenine), and returned to ambient temperature (24°C). RNA was modified by adding 5 μL of freshly made SHAPE reagent, 5 mg/mL 1M7 (1-methyl-7-nitroisatoic anhydride) dissolved in anhydrous DMSO. Modification reactions were incubated at ambient temperature for 12 min and then quenched by 5 μl of 0.5 M Na-MES pH 6.0. Quenches also included 1 μL of poly(dT) magnetic beads (Ambion) and 0.065 pmol of FAM-labeled Tail2-A20 primer for reverse transcription. Samples were separated using magnetic stands, washed thoroughly with 70% ethanol, and air-dried. Beads were resuspended in 5.0 μL reverse transcription mix with SuperScript III (Thermo Fisher), then incubated at 48°C for 30 min. RNAs were degraded by adding 5 μL 0.4 M NaOH and incubating at 90°C for 3 min. Solutions were cooled down on ice then neutralized with 3 μL acid quench (1.4 M NaCl, 0.6 M HCl, and 1.3 M Na-acetate). Fluorescent labeled cDNA was recovered by magnetic bead separation, rinsed 70% ethanol, and air-dried. The beads were resuspended in 10 μL Hi-Di formamide (Applied Biosystems) with 0.0625 μL ROX-350 ladder (Applied Biosystems) and eluted for 20 min. The eluants were loaded onto capillary electrophoresis sequencers (ABI3100 or ABI3730).

Multi-probe chemical mapping was performed with procedures similar to SHAPE chemical mapping, with variations in the modification and quench steps. Specific preparations were as follows: 1% dimethyl sulfate (DMS), mixing 1 μL 10.5 M DMS into 9 μL ethanol, and then 90 μL doubly deionized water ($ddH_2O$); 0.4% glyoxal, dilution to 1/100x of 8.8 M glyoxal; RNase V1, serial dilution to 1/1000x in storage buffer (50% glycerol, 50 mM Tris-HCl pH 7.4, 100 mM NaCl, 0.1 mM EDTA); Terbium(III), 4 mM $TbCl_3$ in $ddH_2O$; FMN, 2 mM flavin mononucleotide in $ddH_2O$. Volumes of 5 μL of these modifier stocks were added to folded RNA solution for 12 min, except for UV treatment, in which samples were exposed directly under a hand-held 302 nm UV lamp for 3 min. FMN photo oxidation reactions were placed on a visible-light box during the entire 12 min reaction. Modifications were quenched by 5 μL of 2-mercaptoethanol for 1M7, DMS, glyoxal and RNase V1; 72 mM EDTA for Terbium(III); or $ddH_2O$ (and removal of light source) for 'no modification' controls, FMN, and UV.

## CE data processing

The HiTRACE 2.0 software was used to analyze CE (capillary electrophoresis) data (*Kim et al., 2013*; *Yoon et al., 2011*; *Lee et al., 2015*). Electrophoretic traces were aligned and baseline subtracted using linear and non-linear alignment routines as previously described (*Kladwang et al., 2014*). Sequence assignment was accomplished semi-automatically with human supervision. Band intensities were obtained by fitting profiles to Gaussian peaks and integrating. Normalization, correction for signal attenuation, and background subtraction were enabled by inclusion of referencing hairpin loop residues (GAGUA) at both 5′ and 3′ ends, 10x dilution replicates, and no-modification controls. Briefly, true values for saturated peaks were obtained from 10x dilutions. Signal attenuation was corrected from 5′ to 3′ ends based on the relative reactivity between 5′ and 3′ referencing hairpin loop intensities (*Kladwang et al., 2014*). Reactivities of SHAPE profiles were normalized against GAGUA, while other modifiers in multi-probe mapping were normalized to subsets of GAGUA which are reactive to that particular modifier.

## Data deposition

All chemical mapping datasets, including $M^2$, $M^2R$, $LM^2R$ and multi-probe mapping, have been deposited at the RNA Mapping Database (http://rmdb.stanford.edu) (*Cordero et al., 2012a*) under the following accession codes: ADD140_1M7_0001, ADD140_1M7_0002, ADD140_1M7_0003, ADD140_1M7_0004, ADD140_1M7_0005, ADD140_1M7_0006, ADD140_1M7_0007, ADD140_1M7_0008, ADD140_1M7_0009, ADD140_1M7_0010, ADD140_1M7_0011, ADD140_1M7_0012, ADD140_1M7_0013, ADD140_1M7_0014, ADD140_RSQ_0001, ADD140_RSQ_0002, ADD140_RSQ_0003, ADD140_RSQ_0004, ADD140_LCK_0001, ADD140_LCK_0002, ADD140_LCK_0003, ADD140_LCK_0004, ADD140_LCK_0005, ADD140_LCK_0006, ADD71_STD_0001, ADD128_STD_0001, ADD140_STD_0001,

ADD140_DCP_0001, ADD140_DCP_0002, RNAPZ5_RSQ_0001, TRP4P6_RSQ_0001, 16SFWJ_RSQ_0001.

## Simulations to infer helix frequencies and correlation values to rescue factors

All RNA families with length between 100 and 250 nt were screened from the Rfam database (http://rfam.xfam.org/) 11.0 FASTA file (Burge et al., 2013). A sequence from each family was randomly picked as representative, and its in silico base pair probability (BPP) matrix was simulated. Base pairing positions to be tested by M$^2$R were selected from this BPP matrix, for BPP values greater than or equal to 1% over any candidate helices that could form 3 Watson-Crick pairs or more. (The 1% BPP cutoff conveys our knowledge that these candidate helices are likely to have finite but potentially low frequency, based on prior literature analysis or experiments; changing the cutoff to lower values such as 0.1% gave similar results and do not change conclusions of the manuscript). For each helix, the rescue factor metric (Equation 4), averaged over simulated M$^2$R quartets for each base pair in the helix, and the mean BPP across those base pairs were calculated. The MATLAB source code for Rfam simulation and rescue factor estimates are available in a GitHub repository: https://github.com/DasLab/m2r_simulations/. To generate a posterior distribution over helix frequencies $F$ given an experimental rescue factor for a helix, all RFAM samples were collected with the same number of base pairs as the candidate helix and simulated rescue factor in the same bin as the experimental rescue factor (binwidth of 0.05 was chosen to allow for sufficient sampling). The helix frequencies $F$(helix) of these samples (as shown in Figures 4 and 6) were visualized as the posterior distribution through a kernel-density estimate (with the ksdensity function in MATLAB). The representative helix frequency (presented in Table 1 and in main text) was taken as the median helix frequency of these RFAM samples.

Evaluating correlation between two helices requires determining the posterior distribution over the correlation value $g(\mathrm{helix1}, \mathrm{helix2})$, given measurements of $F(\mathrm{helix1})$ and $F(\mathrm{helix1}|\mathrm{helix2})$ (see e.g., Equation 2). Two sets of RFAM helix frequency samples corresponding to the experimental rescue factor for helix 1 in the wild type background and for helix 1 in the lock-helix2 mutational background were collected. The posterior distribution of $g(\mathrm{helix1}, \mathrm{helix2}) = \frac{F(\mathrm{helix1}|\mathrm{helix2})}{F(\mathrm{P1})}$ was then estimated based on taking each helix 1 frequency sampled from the latter set (lock-helix2) and dividing by each helix 1 frequency sampled for the former set (wild type). Combination of posterior distributions for a lock-mutate-map-rescue experiments and its 'flipped' variant (in which helix 1 was locked, and helix 2 subjected to mutate-map-rescue) was carried out by multiplying kernel density estimates (KDE) of the posterior distributions $g(\mathrm{helix1}, \mathrm{helix2})$ for the two separate experiments, assuming a flat prior over correlation values. Changing the bandwidth of the KDE estimates changed posterior support values presented in the main text by less than a factor of 2. For $g(\mathrm{P1}, \mathrm{P4B})$, $g(\mathrm{P2}, \mathrm{P4A})$, and $g(\mathrm{P2}, \mathrm{P4B})$, which discriminated between the MWC and non-MWC models, the combination of posterior distributions was also carried out with exponential fits to the individual posterior distributions; final results for posterior support were the same within a factor of 2.

## Acknowledgements

We thank M Ali, S Doniach, and C VanLang for early discussions and experiments on the add riboswitch. We thank Drs. H Schwalbe and B Fürtig for personal communication regarding temperature compensation and NMR analysis and sharing a manuscript before publication. We acknowledge financial support from NIH R01 GM102519 and R35 GM122579 (to RD).

## Additional information

### Funding

| Funder | Grant reference number | Author |
| --- | --- | --- |
| National Institutes of Health | R01GM102519 | Rhiju Das |
| National Institutes of Health | R35GM122579 | Rhiju Das |

The funders had no role in study design, data collection and interpretation, or the decision to submit the work for publication.

### Author contributions
Siqi Tian, Data curation, Software, Formal analysis, Validation, Investigation, Visualization, Methodology, Writing—original draft; Wipapat Kladwang, Investigation; Rhiju Das, Conceptualization, Formal analysis, Supervision, Funding acquisition, Investigation, Methodology, Project administration, Writing—review and editing

### Author ORCIDs
Siqi Tian ⓘD http://orcid.org/0000-0001-5672-1032
Rhiju Das ⓘD http://orcid.org/0000-0001-7497-0972

### Decision letter and Author response
Decision letter https://doi.org/10.7554/eLife.29602.096
Author response https://doi.org/10.7554/eLife.29602.097

## Additional files

### Supplementary files
• Transparent reporting form
DOI: https://doi.org/10.7554/eLife.29602.026

### Major datasets
The following datasets were generated:

| Author(s) | Year | Dataset title | Dataset URL | Database, license, and accessibility information |
|---|---|---|---|---|
| Tian S, Kladwang W, Das R | 2017 | add 140 M2 WT no ligand | https://rmdb.stanford.edu/detail/ADD140_1M7_0001 | Publicly available at the RNA Mapping Database (accession no. ADD140_1M7_0001) |
| Tian S, Kladwang W, Das R | 2017 | add 140 M2 WT w/ ligand | https://rmdb.stanford.edu/detail/ADD140_1M7_0002 | Publicly available at the RNA Mapping Database (accession no. ADD140_1M7_0002) |
| Tian S, Kladwang W, Das R | 2017 | add 140 M2 A109U no ligand | https://rmdb.stanford.edu/detail/ADD140_1M7_0003 | Publicly available at the RNA Mapping Database (accession no. ADD140_1M7_0003) |
| Tian S, Kladwang W, Das R | 2017 | add 140 M2 A109U w/ ligand | https://rmdb.stanford.edu/detail/ADD140_1M7_0004 | Publicly available at the RNA Mapping Database (accession no. ADD140_1M7_0004) |
| Tian S, Kladwang W, Das R | 2017 | add 140 M2 G78C no ligand | https://rmdb.stanford.edu/detail/ADD140_1M7_0005 | Publicly available at the RNA Mapping Database (accession no. ADD140_1M7_0005) |
| Tian S, Kladwang W, Das R | 2017 | add 140 M2 G78C w/ ligand | https://rmdb.stanford.edu/detail/ADD140_1M7_0006 | Publicly available at the RNA Mapping Database (accession no. ADD140_1M7_0006) |

| Tian S, Kladwang W, Das R | 2017 | add 140 M2 A116U no ligand | https://rmdb.stanford.edu/detail/ADD140_1M7_0007 | Publicly available at the RNA Mapping Database (accession no. ADD140_1M7_0007) |
|---|---|---|---|---|
| Tian S, Kladwang W, Das R | 2017 | add 140 M2 A116U w/ ligand | https://rmdb.stanford.edu/detail/ADD140_1M7_0008 | Publicly available at the RNA Mapping Database (accession no. ADD140_1M7_0008) |
| Tian S, Kladwang W, Das R | 2017 | add 140 M2 G44C no ligand | https://rmdb.stanford.edu/detail/ADD140_1M7_0009 | Publicly available at the RNA Mapping Database (accession no. ADD140_1M7_0009) |
| Tian S, Kladwang W, Das R | 2017 | add 140 M2 G44C w/ ligand | https://rmdb.stanford.edu/detail/ADD140_1M7_0010 | Publicly available at the RNA Mapping Database (accession no. ADD140_1M7_0010) |
| Tian S, Kladwang W, Das R | 2017 | add 140 M2 G37C no ligand | https://rmdb.stanford.edu/detail/ADD140_1M7_0011 | Publicly available at the RNA Mapping Database (accession no. ADD140_1M7_0011) |
| Tian S, Kladwang W, Das R | 2017 | add 140 M2 G37C w/ ligand | https://rmdb.stanford.edu/detail/ADD140_1M7_0012 | Publicly available at the RNA Mapping Database (accession no. ADD140_1M7_0012) |
| Tian S, Kladwang W, Das R | 2017 | add 140 M2 G81C no ligand | https://rmdb.stanford.edu/detail/ADD140_1M7_0013 | Publicly available at the RNA Mapping Database (accession no. ADD140_1M7_0013) |
| Tian S, Kladwang W, Das R | 2017 | add 140 M2 G81C w/ ligand | https://rmdb.stanford.edu/detail/ADD140_1M7_0014 | Publicly available at the RNA Mapping Database (accession no. ADD140_1M7_0014) |
| Tian S, Kladwang W, Das R | 2017 | add 140 M2R 1-bp no ligand | https://rmdb.stanford.edu/detail/ADD140_RSQ_0001 | Publicly available at the RNA Mapping Database (accession no. ADD140_RSQ_0001) |
| Tian S, Kladwang W, Das R | 2017 | add 140 M2R 1-bp w/ ligand | https://rmdb.stanford.edu/detail/ADD140_RSQ_0002 | Publicly available at the RNA Mapping Database (accession no. ADD140_RSQ_0002) |
| Tian S, Kladwang W, Das R | 2017 | add 140 M2R 2-bp no ligand | https://rmdb.stanford.edu/detail/ADD140_RSQ_0003 | Publicly available at the RNA Mapping Database (accession no. ADD140_RSQ_0003) |
| Tian S, Kladwang W, Das R | 2017 | add 140 M2R 2-bp w/ ligand | https://rmdb.stanford.edu/detail/ADD140_RSQ_0004 | Publicly available at the RNA Mapping Database (accession no. ADD140_RSQ_0004) |
| Tian S, Kladwang W, Das R | 2017 | add 140 LM2R lock-P1 no ligand | https://rmdb.stanford.edu/detail/ADD140_LCK_0001 | Publicly available at the RNA Mapping Database (accession no. ADD140_LCK_0001) |

| | | | | |
|---|---|---|---|---|
| Tian S, Kladwang W, Das R | 2017 | add 140 LM2R lock-P2 no ligand | https://rmdb.stanford.edu/detail/ADD140_LCK_0002 | Publicly available at the RNA Mapping Database (accession no. ADD140_LCK_0002) |
| Tian S, Kladwang W, Das R | 2017 | add 140 LM2R lock-P4A no ligand | https://rmdb.stanford.edu/detail/ADD140_LCK_0003 | Publicly available at the RNA Mapping Database (accession no. ADD140_LCK_0003) |
| Tian S, Kladwang W, Das R | 2017 | add 140 LM2R lock-P4B no ligand | https://rmdb.stanford.edu/detail/ADD140_LCK_0004 | Publicly available at the RNA Mapping Database (accession no. ADD140_LCK_0004) |
| Tian S, Kladwang W, Das R | 2017 | add 140 LM2R lock-P1B no ligand | https://rmdb.stanford.edu/detail/ADD140_LCK_0005 | Publicly available at the RNA Mapping Database (accession no. ADD140_LCK_0005) |
| Tian S, Kladwang W, Das R | 2017 | add 140 LM2R mut-P2 no ligand | https://rmdb.stanford.edu/detail/ADD140_LCK_0006 | Publicly available at the RNA Mapping Database (accession no. ADD140_LCK_0006) |
| Tian S, Kladwang W, Das R | 2017 | add 71 1D standard states | https://rmdb.stanford.edu/detail/ADD71_STD_0001 | Publicly available at the RNA Mapping Database (accession no. ADD71_STD_0001) |
| Tian S, Kladwang W, Das R | 2017 | add 128 1D standard states | https://rmdb.stanford.edu/detail/ADD128_STD_0001 | Publicly available at the RNA Mapping Database (accession no. ADD128_STD_0001) |
| Tian S, Kladwang W, Das R | 2017 | add 140 1D standard states | https://rmdb.stanford.edu/detail/ADD140_STD_0001 | Publicly available at the RNA Mapping Database (accession no. ADD140_STD_0001) |
| Tian S, Kladwang W, Das R | 2017 | add 140 deep-chemical profiling no ligand | https://rmdb.stanford.edu/detail/ADD140_DCP_0001 | Publicly available at the RNA Mapping Database (accession no. ADD140_DCP_0001) |
| Tian S, Kladwang W, Das R | 2017 | add 140 deep-chemical profiling w/ ligand | https://rmdb.stanford.edu/detail/ADD140_DCP_0002 | Publicly available at the RNA Mapping Database (accession no. ADD140_DCP_0002) |
| Tian S, Kladwang W, Das R | 2017 | GIR1 ribozyme M2R | https://rmdb.stanford.edu/detail/RNAPZ5_RSQ_0001 | Publicly available at the RNA Mapping Database (accession no. RNAPZ5_RSQ_0001) |
| Tian S, Kladwang W, Das R | 2017 | P4P6 domain M2R | https://rmdb.stanford.edu/detail/TRP4P6_RSQ_0001 | Publicly available at the RNA Mapping Database (accession no. TRP4P6_RSQ_0001) |
| Tian S, Kladwang W, Das R | 2017 | 16S FWJ domain M2R | https://rmdb.stanford.edu/detail/16SFWJ_RSQ_0001 | Publicly available at the RNA Mapping Database (accession no. 16SFWJ_RSQ_0001) |

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

## Appendix 1

DOI: https://doi.org/10.7554/eLife.29602.027

## Appendix results

### Conventional SHAPE and mutate-and-map measurements

Before carrying out extensive mutagenesis and compensatory rescue experiments, we tested whether one-dimensional SHAPE profiling on the WT RNA might discriminate between MWC and non-MWC models (main text *Figure 1*). In the absence of adenine, the *add* riboswitch SHAPE profile (main text *Figure 1—figure supplement 1*) showed protections and exposures throughout the aptamer region (nts 14–80), and incubation with 5 mM adenine gave reduced reactivity throughout this region, consistent with formation of P1, P2, and P3, and RNA tertiary structure formation at and around the adenine binding pocket, as expected for the *holo* state. In addition, weak adenine-dependent increases in SHAPE reactivity in the expression platform, the Shine-Dalgarno ribosome binding site and the AUG codon (nts 112–122), supported the adenine-coupled opening of the expression platform that is the hallmark of riboswitch function (main text *Figure 1—figure supplement 1C*). In between these two regions, nucleotides 105–111 also modestly increased in SHAPE reactivity upon adenine binding, although, in both the MWC and non-MWC models, this segment is predicted to be sequestered into a P5 helix in all states (main text *Figure 1A and B*). However, this observation can be accommodated into both the MWC and non-MWC models by assuming partial opening of P5 in the adenine-bound *holo* state (see Discussion) and therefore does not discriminate between the models.

Overall, while consistent with available models, these SHAPE data do not discriminate between them due to difficulties in interpreting these data: changes in SHAPE reactivity do not necessarily reflect changes in base pairing frequencies. Nucleotides that do not form Watson-Crick pairs can appear protected or reactive depending on changes in their local environment. Conversely, nucleotides that form Watson-Crick pairs can appear partially reactive, especially if the pairs are at the edges of helices. For these reasons, we turned to mutate-and-map measurements (described next) and higher-dimensional techniques (compensatory rescue; described in the main text) that do not require detailed interpretations of the absolute reactivity values at particular nucleotides, but rather changes in reactivities integrated across all nucleotides.

Prior to rescue experiments, which involve double compensatory mutants, we systematically scanned single mutants of each nucleotide to its complement, followed by SHAPE profiling across the riboswitch, which gave the two-dimensional patterns in main text *Figure 1—figure supplement 2*. Automated analyses of these mutate-and-map (M$^2$) data to determine the dominant secondary structure and to infer an approximate structural ensemble supported both the MWC and non-MWC models but did not carry the precision to discriminate between them (main text *Figure 1—figure supplement 3*).

### Mutate-map-rescue (M$^2$R) – application generality and automated classification

Our prior work on mutate-map-rescue (M$^2$R) tested RNA structures by disrupting a base pair by single mutation, and testing whether compensatory double mutations restore such perturbations, but relied on manual inspection to assess rescue, and focused on a single model system, a four-way junction domain of *E. coli* 16S rRNA (*Tian and Das, 2016*). We first sought to assess the generality of M$^2$R approach by applying it to other RNA elements. First, we deployed M$^2$R on the lariat-capping GIR1 ribozyme, (*Meyer et al., 2014*) to test helix P5 from crystallographic model against its alternative version (alt-P5), which was predicted by our M$^2$-based model with a 2-register shift (*Appendix 1—figure 2*). For each base pair tested, we compared the SHAPE profiles of WT (wild type), two single mutants, and the double mutant (rescue) in a 'quartet'. With the local perturbations caused by single mutants in the P5/alt-P5

region, double mutants based on the P5 successfully restored the SHAPE profile to WT, while double mutants based on the alt-P5 failed to rescue. Thus, M²R provided evidence for the P5 helix but not the register-shifted alt-P5, consistent with the crystallographic structure of the ribozyme. In addition, we applied M²R on the P4-P6 domain of the *Tetrahymena* ribozyme (*Cate et al., 1996*) to test for presence of crystallographic P5c against an alternative suggested by mutate-and-map analysis (*Appendix 1—figure 2*). Finally, M²R-validated secondary structure models were confirmed experimentally on two RNA domains by functional assays in vitro or *in cellulo*: a human *HoxA9* mRNA IRES (internal ribosomal entry site) domain (*Xue et al., 2015*) and a stem loop domain from Influenza A virus (PSL2IAV_RSQ_0001 at the RNA mapping database, https://rmdb.stanford.edu/detail/PSL2IAV_RSQ_0001).

**Appendix 1—figure 1.** Base-pair-wise classification of M²R and LM²R quartets by the 4-bin auto-score. Table summary of 4-bin classification by human expert and automatic algorithm on in vitro (**A**) single base-pair M²R, (**B**) double base-pair M²R, and (**C**) LM²R and MutP2 are shown. Each symbol represents one quartet testing a single base-pair (or two adjacent base-pairs in (**B**)).
DOI: https://doi.org/10.7554/eLife.29602.028

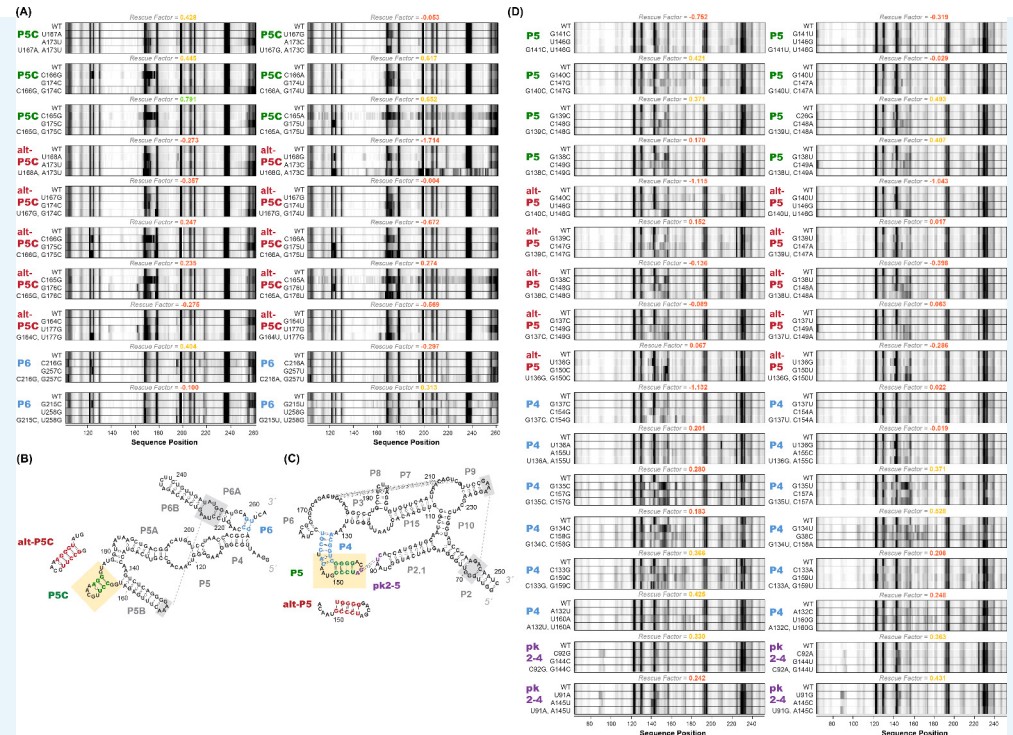

**Appendix 1—figure 2.** M²R quartets and helices tested for P4-P6 and GIR1. (**A**) M²R quartets of P4-P6 domain testing P5C, alt-P5C, and P6. Rescue factor for each quartet is given in the title and colored as in Figure S1. (**B–C**) Secondary structures of P4-P6 and GIR1 highlighting tested helices. (**D**) M²R quartets of GIR1 ribozyme testing P5, alt-P5, P4, and pk2-5.
DOI: https://doi.org/10.7554/eLife.29602.029

In order to evaluate the accuracy of M²R on a larger scale, we turned to in silico simulation to generate a much greater number of RNA test cases. Specifically, we asked: do M²R-validated base pairs have high base-pairing probability (BPP) in silico? To answer this question, we sought to automate the calling of M²R result with an automated classifier that would evaluate M²R data with results matching human inspection, and this study eventually led to the definition of the 'rescue factor' applied to the adenine riboswitch. Our first scoring function took into account the following two factors: 1) the amount of perturbation seen in each of the two single mutants, compared to wild type profiles; and 2) the similarity (rescuing effect) between double mutant and WT. We captured the latter effect through a rescue factor metric:

$$\text{Rescue factor} \equiv 1.0 \; - - \frac{RMSD(\text{WT}, \text{AB})}{\max(RMSD(\text{WT}, \text{A}), \; RMSD(\text{WT}, \text{B}))}$$

where

$$RMSD(\text{A}, \text{B}) = \sqrt{\frac{1}{N} \sum_{i=1}^{N} [d_A(i) - d_B(i)]^2}$$

and $d_A$ and $d_B$ are vectors of normalized SHAPE profiles, with the same length of $N$ (see Appendix Methods for further details). The classifier returned a result from three categories: Validated, Falsified, or Uncertain; the last one was assigned for cases where either (1) single mutants failed to introduce discernible perturbations, or (2) the rescuing effect was 'half-way' and hard to assign. After training with the in vitro M²R data on the GIR1, P4-P6, and IRES measurements as well as in silico simulated counterparts, we obtained a classifier that recovered human expert calls (*Appendix 1—figure 2*).

Next, we simulated M²R in silico on 325 RNA families from the Rfam database (*Burge et al., 2013*) whose lengths were in range between 100 and 250 nt. More than 37,000

base pairs were tested from helices that were longer than 2 bp and with predicted base pairing probability (BPP) greater than 1%. *Appendix 1—figure 3* shows the BPP distribution of the 4-category classification. These results confirmed that the Validated and Falsified categories corresponded to base pairs with high and low BPP. Unexpectedly, we also observed a quantitative correlation between the in silico predicted BPP and the 'rescue factor' metric that underlies our classification, even for base pairs that appear with BPP frequencies between 0.3 and 0.7 (main text *Figure 2C*). Furthermore, for helices of longer length, this relationship became more well-defined. We reasoned that this correlation would enable estimation of BPP based on helix length and the 'rescue factor' metrics measured and averaged over all the base pairs for a given helix (*Appendix 1—figure 3*), and that this quantitative metric would be more informative than a qualitative 4-category classification. Further support for using the 'rescue factor' to estimate a posterior probability for tested helix frequencies arose from our studies on the adenine riboswitch with and without adenine, which could be compared to 'gold standard' helix frequency estimates from NMR analysis, as described in the main text.

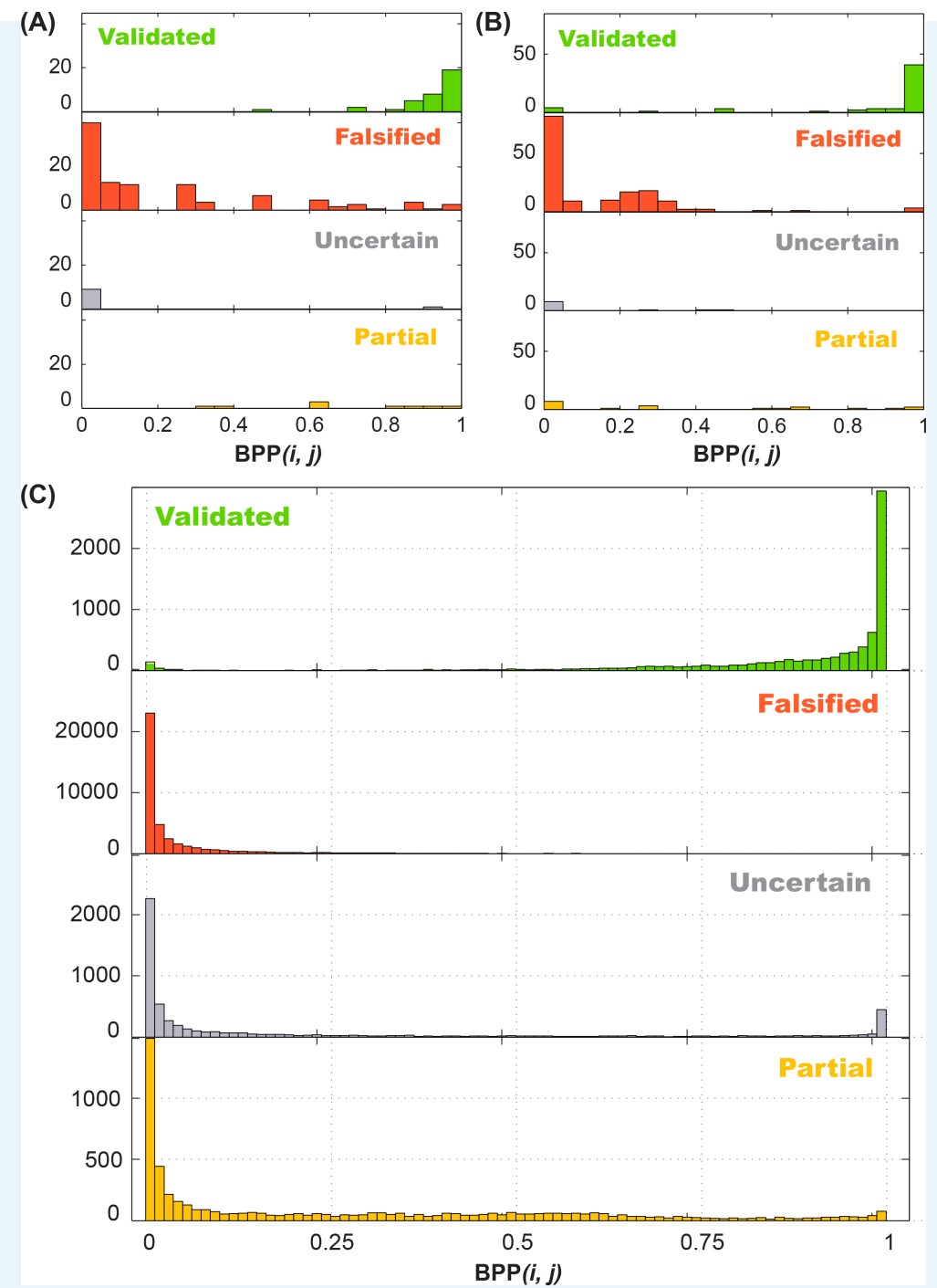

**Appendix 1—figure 3.** Performance of 4-bin auto-score classifier. Histogram of in silico (**A**) training data (16S-FWJ, P4P6, GIR1, Hox, PB2; total of 162 quartets), (**B**) test data (*add* single and double base-pair M$^2$R; total of 242 quartets), and (**C**) Rfam family (total of 62484 quartets) simulations are shown by their classification, and grouped by the in silico predicted base-pairing probability (BPP).

DOI: https://doi.org/10.7554/eLife.29602.030

## Current limitations of M$^2$R and classifier

We trained an automated classifier based on previous M$^2$R in vitro data combined with in silico simulations. The classifier was later independently tested on *add* single-base-pair M$^2$R, double-base-pair M$^2$R, and LM$^2$R datasets, showing agreement with calling by an expert human referee

(S.T.) (*Appendix 1—figure 1*). Moreover, the BPP distribution of the simulated in silico counterparts for *add* riboswitch follows the same trend as Rfam tests, in agreement with its generality. The performance of such a simple classifier was acceptable and supported our heuristic for the rescue factor (*Appendix 1—figure 3*). Our preliminary attempt on 4-category classification gave conservative conclusions for M$^2$R quartets, with 'Partially Validated' as weaker cases for incomplete compensatory rescue. For final analyses, we chose to present final rescue factors and corresponding helix frequencies (see, e.g., main text *Figures 3* and *6*); the classifier based on the rescue factor gives a more qualitative picture of the results, but allows simple visual checks that for each helix, different base pairs tested for compensatory rescue give concordant 'calls' for the frequency of the helix (*Appendix 1—figure 1*).

## Adenine riboswitch construct design

We performed the M$^2$R pipeline on a 128-nt *add* riboswitch construct, which has 15 extra nucleotides on the 3′ end into the coding sequence compared to a 112-nt one used in a recent study (*Reining et al., 2013*). We chose this longer *add* version after reproducing prior work in our laboratory (*Ali, 2011*; *Cordero and Das, 2015*) finding that the sequence context has an effect on the folding landscape of this riboswitch (main text *Figure 1—figure supplement 1*). Although the 112-nt construct showed weak ligand-responsive chemical reactivity changes in the SD region under the salt conditions used in NMR (*Cordero and Das, 2015*), it does not show noticeable switching under our in vitro solution conditions. The extended construct gives chemical reactivity changes under all conditions tested.

## Multi-probe chemical mapping tests partitioning predicted by the MWC model

As discussed in the main text introduction, the MWC model requires that, even in the absence of adenine ligand, the *add* riboswitch samples two states: *apoA* and *apoB*, that have secondary structures compatible and incompatible with aptamer adenine binding, respectively, and that these states are more likely or less likely, respectively, to have free ribosome binding sites. We tested this model with the aid of the lock-P1 and lock-P4A mutants, which isolated the *apoA* and *apoB* states, respectively, and chemical mapping to monitor the riboswitch structural ensemble at nucleotide resolution. To stringently test the prediction, we expanded our chemical mapping protocol to include five additional modifiers beyond SHAPE: DMS (dimethyl sulfate) (*Cordero et al., 2012b*), glyoxal (*Banerjee et al., 1993*), RNase V1 (*Zou and Ouyang, 2015*), terbium(III) (*Hargittai and Musier-Forsyth, 2000*; *Harris and Walter, 2003*), and ultraviolet irradiation at 302 nm (*Kladwang et al., 2012*), giving a total number of 1920 measurements measured in three sequences that would test a two-state partitioning. As expected from the model of *apoA* vs. *apoB*, these multiple modifier profiles varied significantly between the lock-P1 and lock-P4A mutants, especially in the expression platform regions (see, e.g., nts 98–112). Nevertheless, as predicted from the MWC model, a simple linear combination of the profiles, assuming 48% of the *apoA* state and 52% of the *apoB* state recovered the data measured for the adenine-free wild type riboswitch within experimental errors, except at nucleotides 79–81 (main text *Figure 5—figure supplement 1A–B*). We note that there was only one parameter optimized in this fit; scaling of each data profile was carried out based on well-defined flanking hairpins as normalization standards (*Kladwang et al., 2014*). These values for the *apoA* vs. *apoB* state frequencies agree within errors with helix frequencies measured above in compensatory rescue experiments as well as with the prior NMR experiment at similar temperature (40% *apoA* at 30°C) (*Reining et al., 2013*). However, the discrepancies at nucleotides 79–81 were reproducible, suggesting that there are additional secondary structures, potentially many at low frequencies, beyond those detected here in the ligand-free ensembles.

# Appendix methods

## In silico partition, autoscore classifier and Rfam sampling

In silico RNA SHAPE profiles were simulated using the partition executable from the RNAstructure package version 5.6 (*Mathews et al., 2004*; *Reuter and Mathews, 2010*). The resulting pair-wise probability matrices was projected into a one-dimensional vector to get per-residue base pairing probabilities (BPP). For $M^2R$ quartet simulations, the targeted base pair was mutated to G-C or C-G pairs (i.e., the 'Stable' library from Primerize-2D [*Tian and Das, 2017b*]). Flanking sequences (5′ or 3′) were not included for simulation.

The autoscore classifier utilizes a 'ratio' metric for classification. Normalized SHAPE profiles are used as input. First, the difference between two SHAPE profiles are calculated by:

$$RMSD(A, B) = \sqrt{\frac{1}{N}\sum_{i=1}^{N}[d_A(i) - d_B(i)]^2}$$

Where $d_A$ and $d_B$ are vectors of normalized SHAPE reactivity values, with the same length of $N$.

For a given quartet of SHAPE profiles (WT, A, B, AB), we first determined whether there was pronounced perturbation introduced by either single mutants (A and B) by calculating:

$$\max(RMSD(WT, A),\ RMSD(WT, B)) \tag{A1}$$

If the value of (1) is less than a *CUTOFF*, the Uncertain class is assigned. Otherwise, continue to determine the amount of rescuing effect by the double mutant (AB):

$$\frac{RMSD(WT, AB)}{\max(RMSD(WT, A),\ RMSD(WT, B))} \tag{A2}$$

[2] defines the 'ratio' metric, that is, the ratio between the distance of AB to WT and the maximum distance of A or B to WT. If the value of [2] is less than *LOW*, the Validated class is assigned due to the relatively small residual of difference of AB to WT compared to single mutants. If the value of (2) is greater than *HIGH*, the Falsified class is assigned. For values of (2) between *LOW* and *HIGH*, the Partial class is assigned, which captures cases that show incomplete rescue or a mixed result.

The classifier was trained on in silico and in vitro data of GIR1, P4P6, 16S-FWJ, HoxA9 and PSL2 (PSL2IAV_RSQ_0001 at the RNA mapping database, https://rmdb.stanford.edu/detail/PSL2IAV_RSQ_0001), a total of 324 quartets. Data were cropped to the region of interest (i.e., excluding the flanking sequences); in silico simulated reactivity profiles were used directly, while in vitro data were attenuation-corrected and normalized by internal standards (*Kladwang et al., 2014*). For each quarter, a manual score in 1 of the 4 categories was available from expert inspection (by S.T.). The set of parameters *CUTOFF* (0.1), *LOW* (0.4), and *HIGH* (0.7) were then determined by grid search, optimizing for recovery of the expert assessments. The classifier was then tested on in silico and in vitro data for the *add* riboswitch using the chosen parameters. Both training and test data compose of Falsified vs. Validated cases in a ~2.8:1 ratio.

## Structural equilibrium fitting

Equilibrium fractions of each state were determined by assuming that lock-P1 and lock-P4A stabilizers completely stabilize the corresponding structure – their reactivity profiles therefore represent the reactivity profile for each state. The reactivities of lock-P1 and lock-P4A were taken to fit the WT reactivity by $\chi^2$ score, which is calculated as follows:

$$\chi^2 = \sum_i \frac{(d_{WT} - d_{PRED})^2}{\sigma_{WT}^2 + \sigma_{PRED}^2}$$

$$d_{PRED} = \alpha \cdot d_{apoA} + (1-\alpha) \cdot d_{apoB}$$

$$\sigma^2_{PRED} = \alpha^2 \cdot \sigma^2_{apoA} + (1-\alpha)^2 \cdot \sigma^2_{apoB}$$

Where $d_{WT}$, $d_{apoA}$, and $d_{apoB}$ are mean SHAPE reactivity profiles of WT, lock-P1, and lock-P4A, and $\sigma_{WT}$, $\sigma_{apoA}$, and $\sigma_{apoB}$ are errors (standard deviation) of $d_{WT}$, $d_{apoA}$, and $d_{apoB}$. The parameter $\alpha$ is the fraction of the *apoA* state, ranging from 0 to 1. $\chi^2$ is summed over all or a subset of nucleotide positions *i* as specified below. $\chi^2$ scores are plotted against $\alpha$, and the $\alpha$ value with minimum $\chi^2$ is taken as the best fit. To prevent a single nucleotide position from dominating the fit by their extreme values, nucleotides 95 (SHAPE), 66 (DMS), and 85 (glyoxal) were excluded.

