## [Decision Letter]

Thank you for submitting your article "Allosteric Logic of the *V. vulnificus* Adenine Riboswitch Resolved by Four-dimensional Chemical Mapping" for consideration by *eLife*. Your article has been favorably evaluated by James Manley (Senior Editor) and three reviewers, one of whom, Anna Marie Pyle (Reviewer #3), is a member of our Board of Reviewing Editors. The following individuals involved in review of your submission have agreed to reveal their identity: Alain Laederach (Reviewer #2); Philip Bevilacqua (Reviewer #4).

The reviewers have discussed the reviews with one another and the Reviewing Editor has drafted this decision to help you prepare a revised submission.

Summary:

All three reviewers were enthusiastic about the quality and significance of the work, but there were also serious concerns about the clarity and presentation of the manuscript. As written, reviewers felt that the paper would lack the impact that it deserves and made numerous suggestions for rewriting. Several important scientific issues were also raised and these will need to be rigorously addressed. With these changes and modifications, the reviewers are confident that this will be a high impact paper of major significance to the field of RNA structure and folding.

Essential revisions:

Each reviewer has indicated specific problems with the writing and presentation and these will all need to be addressed. The revised manuscript must be clear and comprehensible to the RNA community. There are specific scientific criticisms, such as point 8 from reviewer 3 that must be addressed in the revised text as well.

*Reviewer #2:*

In the manuscript by Tian and coworkers the authors analyze a remarkable structural data set they collected on the *V. vulnificus* riboswitch to address a very basic question about the underlying allostery governing its main mechanism of action. The precise question being interrogated is whether a population shift model really explains RNA allostery is this model or whether an alternative model is better as has been suggested in other studies. The resulting data, based on "locked ensemble" analyses demonstrated with the data in Figure 4 provides compelling evidence for the MWC model. Furthermore, the approach here is a completely novel application of SHAPE probing combined with mutation and mapping to answer a very fundamental biophysical question about RNA folding.

My one criticism of the paper, and I believe this is just a presentation issue is on the biological significance and generalizability of these results. I believe the authors could do a better job of explaining to the more biologically oriented reader why it is important to the function of RNA in the cell why RNA using a population shift model is important. Furthermore, I think the authors should take the time to (re)-explain why understanding folding in this system is "paradigmatic" and what the possible biological ramifications are of understanding the folding of the *add* riboswitch at this level of detail. As such I recommend the following major revisions:

1) I think the major reasoning for this study needs to be better illustrated and explained. It is important to remind the reader that the *add* riboswitch being studied is the aptamer domain of a regulatory switch and that allosteric interactions are likely key to the regulatory mechanism. Can the authors explain how functional regulation might differ under both folding models, perhaps even predict how each model would impact the kinetics? I think some kind of "cartoon" figure would go a long way to explaining this.

2) Similarly I think a workflow figure of some type is needed to explain how studying each "locked ensemble" can then be reintegrated into a more general understanding of allostery and folding mechanism. I hate to ask for more cartoons, but the authors need to somehow make this concept clearer in their presentation of their work.

3) More fundamentally there are some underlying assumptions here that it is possible to lock conformations. The design strategy for these different molecules needs to be better explained as well as the rationale for then being able to reintegrate each individual molecule’s unique folding properties into a general model of WT folding.

4) As impressive as the number and meta-analysis of all the mutate and map data is, I think the authors could move a lot of the data they show in Figure 3 to the supplement to focus more on the concepts and their biophysical interpretation.

*Reviewer #3:*

In this paper, the authors utilize an expanded form of the mutate and map procedure to dissect intermediate states involved in folding and activation of a riboswitch and to resolve controversies about the structural states that are actually visited in the ligand-free state. This is an area of intense interest because structural studies on riboswitches conducted to date do not actually explain how the riboswitch is kept OFF, as the ligand-free forms of these molecules can sample structural states that still expose the ribosome binding. This is an important paper and a rigorous study that exemplifies the power of RNA mapping studies for quantitative analysis of RNA folding.

Unfortunately, the paper suffers from two major flaws that prevent the authors from effectively making what are important points for the community.

1) This paper gets in its own way: It is written in confusing, convoluted language and important points are made using terms that are undefined and will be opaque to much of the RNA structural biology community. The authors are attempting to convey complex ideas in this paper, using a new approach that is inherently complex. This is an impossible task without clear, precise language and terminology.

2) The authors fail to utilize their figures as display items in support of their ideas. In the text, the authors make specific claims about their data, referring to complicated figures that lack clear notation *on the figure* to direct the reader to the actual result. Even in cases where the figures are intuitively interpretable, effects are small and therefore it is essential that the authors explain the figures more carefully, indicating exactly what they mean in the text.

Specific additional comments are below:

1) In the Introduction, the language is quite opaque, which makes an already complicated system even more difficult to understand. Example: "ligand binding to an aptamer region of the RNA shifts a pre- existing equilibrium between strikingly different secondary structures, resulting in allosteric exposure or sequestration of another sequence-distal region harboring a ribosome binding site or other gene expression platform (Fernández-Luna and Miranda-Ríos, 2008; Gilbert and Batey, 2006; Breaker, 2012)." Even sophisticated readers will not know what "allosteric exposure" or "sequestration" mean in this context. The authors are actually explaining a simple phenomenon, but they need to use clear language that is not cluttered with jargon and terms with multiple meanings. Interestingly, and noted later in this review, the goals of this paper are clearly articulated within the Results section, suggesting that a rearrangement of the text and of the ideas might result in greater clarity (see point 6, below).

2) In paragraph 2 of the Introduction, in part due to the problem stated above, the authors have not clearly articulated the discrepancy between models, nor have they defined what exactly is meant by "allosteric logic".

3) Again, the authors have not explained their premise well: "Briefly, if mutation of either side of a base pair (mutant A or mutant B) disrupts the reactivities at other nucleotides in the RNA, but the compensatory double mutation (mutant AB) restores the SHAPE reactivities to the wild type (WT) profile, these observations suggest that the two sites were base paired in the starting WT sequence. Our prior work assumed that such observed compensation would arise with negligible probability if the mutated nucleotides were not paired." This reader was completely confused by this statement. What is different from the assumption in the prior work? I think the meaning got lost in the double negative describing the prior work, but I cannot be sure what he means.

4) Subsection “Inferring helix frequencies through quantitative compensatory rescue”. Please clearly define exactly what is meant by "helix frequency" in the text. On what experimental metric is it deduced and how is it calculated? If it is in the Materials and methods, then refer to that in the Results section "Helix frequency is defined as XXX and it was calculated as described in Materials and methods".

5) "[…] median values of the helix frequency posterior probability distributions". What is the meaning of "posterior" in this context? Helix frequency distribution has not even been clearly defined yet – so it is vexing that there is now a posterior form of it.

6) Subsection “The *add* riboswitch system”, first two paragraphs. In the Results, the authors finally articulate the structural issues that are being examined in the paper. Almost the entire content of those paragraphs should be used as the Introduction, rather than the present musing on allostery. For example, the reader needs to learn that "The major uncertainty regards what happens when adenine is not present but the RNA nevertheless samples a secondary structure poised to bind adenine, with P1, P2, and P3 formed. This state is termed apoA." It would be clearer to emphasize this right from the beginning, rather than initiating the paper with a treatise on allostery.

7) "Unexpectedly, in between these two regions, nucleotides 105-111 also modestly increased in SHAPE reactivity upon adenine binding, although, in both the MWC and non-MWC models, this segment is predicted to be sequestered into a P5 helix in all states (Figure 2)." This sounds really interesting, but if one looks closely at the figure and lines up nucleotides 105-111, the effect being referred to is small, and potentially attributable to partial stacking of this region near the helix rather than helical formation itself. This reviewer is concerned that the authors are overinterpreting the small effects that are side-effects of the SHAPE method itself (see point 8).

8) A general criticism of the science rather than the writing: A potential flaw in the analysis is the strict interpretation of SHAPE signals in this analysis: The authors assume that shape data can be interpreted in a strictly binary way – representing a population of paired or unpaired duplexes. But SHAPE reactivity is not strictly indicative of base pairing and instead reflects the relative "floppiness" of a 2'-OH – not whether it is paired or not. As a result, there can be partially ordered conformations, and conformationally restricted conformations of a motif that is not double stranded yet can give SHAPE protections of varying magnitude. These issues raise concern about the focus on subtle effects being studied by the investigators.

9) Another result that sounds really interesting but is difficult to discern from the data presented: "Compared to the WT sequence, many of these mutants gave protections or enhancements in the 5 ´-most aptamer region of the riboswitch (nts 25-32) and concomitant exposures or protections (respectively) near the 3 ´ end of the riboswitch (nts 105- 122), which included both the start codon and Shine-Dalgarno ribosome binding sequence (arrows, Figure 3)." Exactly where in the figures is this shown? It must be indicated on the display item.

10) Figure 3–Figure 5: Please define "cumulative posterior possibility" in the Results portion of the text (or better yet, come up with a term that is simpler), as an understanding of the parameter is a prerequisite for analyzing the data presented.

11) Figure 5 and throughout text: If the MWC model is correct, then why isn't the riboswitch constitutively ON 40% of the time in the absence of adenine? Or is the level of "leaky" expression commensurate with the authors' estimates of the population of this state?

12) "Similarly, compensatory rescue targeting P2 gave a significant helix frequency (42%) when P1 was locked (Figure 4, Figure 4—figure supplement 2, and Table 1). As a further check, both models under question predicted the absence of P4A in apoA; indeed, in either the lock-P1 or lock-P2 background, we did not detect compensatory rescue targeting P4A, and could set an upper bound on its frequency of 4%. Finally, we tested the helix frequency of P4B, which was predicted to be absent (0%) in apoA by the MWC model and present (100%) in the non-MWC model (Figure 2)." This is interesting, but the authors do not indicate exactly which parts of Figure 4 support this model. They simply refer to Figure 4, which is a complex figure lacking any direction to the reader. One cannot evaluate the veracity of the authors' claim.

13) Last paragraph of the Results: Again, the authors need to refer directly to the features in Figure 4 that support the helix frequencies being described in the text, and presumably graphed in parts E and F. When inspected by this reviewer, the display items in Figure 4 did not clearly indicate the results claimed. They need to be indicated on the display and referred to in the text.

14) First paragraph of the Discussion: "Four-dimensional chemical mapping offers an experimental route to this information that is unique in its high throughput and use of standard equipment (capilly electrophoretic sequencers)." It is clear from the data provided in this paper that this is not a high throughput method, in fact it is quite the contrary. Also: one presumes that the authors mean "capillary". Any reference to this being a simple or high throughput experiment should be removed, as it unfairly inflates expectations within the community of the methods that should be reasonably applied in structural studies any other RNA molecule.

*Reviewer #4:*

Tian and co-workers provide a deep, high-resolution picture of structural transitions in a functional RNA molecule, the adenine riboswitch using a method they pioneered called "lock-mutate-map-rescue". The authors lock various helices in place through mutations and then judge the effects on the remainder of the molecule via HT rescue experiments. The results favor the famous MWC model originally developed to explain allostery in proteins via conformational selection. The authors clearly address work of another study on the subject and the issue of whether in the absence of ligand the binding site is open, and the concern that existing data support both MWC and non-MWC models. The authors' work has been tested statistically using Bayesian statistics. Importantly, they show that conventional 1D SHAPE experiments cannot differentiate between MWC and non-MWC models, motivating their current approach. Mutants are made that stabilize P1 and P2 that are outside of the critical P4 switch helix, and neither favor formation of P4B. And the converse experiment of locking P4B did not form P1 and P2. This led to the main finding, which is that gene expression is ON even in the absence of ligand as long as the aptamer is folded, albeit rarely sans ligand. This is an important and influential finding.

We have two major concerns.

1) At many times the paper is very difficult to follow. Figures are often not described in a way that the reader can access. The language is at times highly specialized and ill defined. If the authors can explain their data in an accessible way, it will greatly increase the impact of the findings.

2) While an elegant method, 4D chemical mapping is not without its concerns. Our only major concern is the general assumption that locking a helix informs on the natural RNA conformational switch. Could locking a helix prevent certain natural pathways that involve transient interactions between helices, such as pseudoknots involving helices themselves, including defects, or could a lock prevent the need to transiently unfold a helix during the switching.

---

## [Author Response]

Reviewer #2:[…] My one criticism of the paper, and I believe this is just a presentation issue is on the biological significance and generalizability of these results. I believe the authors could do a better job of explaining to the more biologically oriented reader why it is important to the function of RNA in the cell why RNA using a population shift model is important.

We agree that the biological significance was not presented clearly. We now state in first paragraph of introduction: “If these multiple states could be characterized in detail and quantitatively modeled, they would offer potential new substrates for biological control and new targets for antibiotic development.” Later in the Introduction, “Understanding the mechanism of allostery is particularly important for developing strategies for controlling riboswitches. For example, potential antibiotics that attempt to constitutively turn on expression of the *add* gene by selectively stabilizing the aptamer secondary structure may not succeed if the MWC model is incorrect”.

Furthermore, I think the authors should take the time to (re)-explain why understanding folding in this system is "paradigmatic" and what the possible biological ramifications are of understanding the folding of the add riboswitch at this level of detail. As such I recommend the following major revisions:1) I think the major reasoning for this study needs to be better illustrated and explained. It is important to remind the reader that the adenine riboswitch being studied is the aptamer domain of a regulatory switch and that allosteric interactions are likely key to the regulatory mechanism. Can the authors explain how functional regulation might differ under both folding models, perhaps even predict how each model would impact the kinetics? I think some kind of "cartoon" figure would go a long way to explaining this.

We now devote more space in the Introduction to explain how the short length and early modeling of the *add* system has made it paradigmatic in studies of RNA conformational change: “A particularly well-studied – but still incompletely understood – example of an allosteric RNA is the *add* riboswitch, a ~120-nucleotide element that resides in the 5´ untranslated region (UTR) of the *add* adenosine deaminase mRNA of human pathogenic bacterium *Vibrio vulnificus*. […] Almost all work on the *add* riboswitch to date have assumed some variation of a Monod-Wyman-Changeux model (MWC, also called

‘conformational selection’ or ‘population shift’ models)”. For related issues on presentation raised by reviewer #3, please see below. For a discussion of kinetic mechanisms (which we cannot currently probe), please see response to reviewer #4 below.

2) Similarly I think a workflow figure of some type is needed to explain how studying each "locked ensemble" can then be reintegrated into a more general understanding of allostery and folding mechanism. I hate to ask for more cartoons, but the authors need to somehow make this concept clearer in their presentation of their work.

We completely agree that the workflow was not clear from the original Figure 1. In fact, in discussions with other colleagues and from these reviews, we realized that a more fundamental point of confusion was in defining the question. We have therefore dramatically changed the presentation to define a single ‘binary’ question addressed in the manuscript: is there strong anticorrelation or correlation between the functional elements at play in the *add* riboswitch.

We have eliminated the prior Figure 1, and indeed any other figures from our original manuscript that show base pairing probability matrices, which are not necessary for understanding the experimental method or the question, and were confusing.

Instead, Figure 1 now shows cartoons of structural ensembles in MWC vs. non-MWC models, and defines the question in terms of a correlation value. The first section of Results is entirely new, and explains why we need to define the question in terms of correlations, and why locking one helix and measuring the frequency change of other helices can answer the question. Again, given the importance of this conceptual background, we give explicit illustrations of the possible models. For example, we have added a paragraph to the first section of results: “More quantitatively, suppose that P1 and P4B arise with 50% frequency in the ligand-free *add* riboswitch structural ensemble. If there is no allosteric ‘communication’ between these two elements, the joint frequency of observing both P1 and P4B should be 50% x 50% = 25% […]”, and this discussion corresponds to new cartoon panels in Figure 1.

3) More fundamentally there are some underlying assumptions here that it is possible to lock conformations. The design strategy for these different molecules needs to be better explained as well as the rationale for then being able to reintegrate each individual molecule’s unique folding properties into a general model of WT folding.

We fully agree – we now give a new main text Figure 5 that shows how different locking mutants for each helix produce SHAPE profiles that are distinct with wild type but similar to each other; this cross-comparison provided important consistency checks that we were locking the targeted helices through independent mutations. Furthermore, we describe two-state fits that demonstrate recovery of wild type SHAPE, DMS, and other data based on mixtures of profiles from the lock mutants. There is now a more careful discussion of these mutants in the Results, in the paragraph that begins with “To identify appropriate locking mutants, we noted that a number of double-base-pair compensatory mutants for P1, P2, P1A, P4A, and P4B (Figure 3—figure supplement 2) exhibited SHAPE profiles that differed from WT RNA but agreed well with each other (Figure 5) […]”.

In particular, we spell out evidence for locking carefully for P1A, P4A, and P4B: “For other helices P1B, P4A, and P4B, we were able to find double-base-pair mutants that gave similar SHAPE profiles to each other, supporting the use of those mutants as helix-locked variants (Figure 5). […] Further evidence that these mutants were appropriately locked came from excellent two-state fits that recovered the adenine-free wild type chemical mapping profiles across five modifiers from the profile for any of these P1B- P4A- or P4B- lock mutants with the profiles for any of the P1 or P2-locking mutants (Figure 5—figure supplement 2).”

In terms of the rationale for how measurements on mutants allow inferences on the wild type folding, we now give two new panels Figure 2 showing how mutate-map-rescue was developed on a prior model system (a four-way junction of the *E. coli* rRNA) and – importantly – cartoons of the structural ensembles that illustrate how results of compensatory mutagenesis provide information on helix frequencies of the wild type.

4) As impressive as the number and meta-analysis of all the mutate and map data is, I think the authors could move a lot of the data they show in Figure 3 to the supplement to focus more on the concepts and their biophysical interpretation.

We agree that those SHAPE and mutate-and-map data are not necessary for the general reader, and distract from our main point – that we can experimentally probe correlations and anticorrelations. We have moved the mutate-and-map data in Figure 3 to supplementary figures for Figure 1. We have moved description of those data to the Supplemental Information. In addition, we have re-focused all main text figures to mainly showing raw data for P1, P4B, and their lock mutants, which illustrate all the relevant measurements and concepts. We believe that this new presentation, combined with new cartoons and better motivation text, is much clearer than the presentation of the original manuscript.

Reviewer #3:[…] Unfortunately, the paper suffers from two major flaws, that prevent the authors from effectively making what are important points for the community.1) This paper gets in its own way: It is written in confusing, convoluted language and important points are made using terms that are undefined and will be opaque to much of the RNA structural biology community. The authors are attempting to convey complex ideas in this paper, using a new approach that is inherently complex. This is an impossible task without clear, precise language and terminology.2) The authors fail to utilize their figures as display items in support of their ideas. In the text, the authors make specific claims about their data, referring to complicated figures that lack clear notation on the figure to direct the reader to the actual result.

We have now gone through all figures and main text to simplify figures and add these kinds of graphical annotations (with big arrows or text labels). Examples include marking “mixed” conformations with red rectangles in Figure 1; “yellow arrow” marking deviations from compensatory rescue in Figure 2; and use of a gold/blue highlighting scheme through all figures to track P1 and P4B as illustration helices.

Even in cases where the figures are intuitively interpretable, effects are small and therefore it is essential that the authors explain the figures more carefully, indicating exactly what they mean in the text.

Similar points were made by reviewers #2 and #4; please see our description of extensive changes in response to both reviewers above and following.

Specific additional comments are below:1) In the Introduction, the language is quite opaque, which makes an already complicated system even more difficult to understand. Example: "ligand binding to an aptamer region of the RNA shifts a pre- existing equilibrium between strikingly different secondary structures, resulting in allosteric exposure or sequestration of another sequence-distal region harboring a ribosome binding site or other gene expression platform (Fernández-Luna and Miranda-Ríos, 2008; Gilbert and Batey, 2006; Breaker, 2012)." Even sophisticated readers will not know what "allosteric exposure" or "sequestration" mean in this context. The authors are actually explaining a simple phenomenon, but they need to use clear language that is not cluttered with jargon and terms with multiple meanings. Interestingly, and noted later in this review, the goals of this paper are clearly articulated within the Results section, suggesting that a rearrangement of the text and of the ideas might result in greater clarity (see point 6, below).

We agree that the Introduction was confusing, introducing too many terms without explanation or figure illustration. We have extensively rewritten the Introduction. For example, we eliminated the sentence pointed out by the reviewers in favor of a more concrete picture of an RNA conformational change in the adenine riboswitch (now presented in Figure 1, rather than being delayed to Figure 2 in the original manuscript).

2) In paragraph 2 of the Introduction, in part due to the problem stated above, the authors have not clearly articulated the discrepancy between models, nor have they defined what exactly is meant by "allosteric logic".

We have now moved significant portions of this more concrete ‘Results’ section up to the Introduction. We have also better spelled out the analogy between the MWC (population shift/conformational selection) mechanism to the classic allosteric mechanism for hemoglobin. We have changed the name ‘allosteric logic’ (which is not really in wide use) to the more conventional ‘allosteric mechanism’ in the title and throughout.

3) Again, the authors have not explained their premise well: "Briefly, if mutation of either side of a base pair (mutant A or mutant B) disrupts the reactivities at other nucleotides in the RNA, but the compensatory double mutation (mutant AB) restores the SHAPE reactivities to the wild type (WT) profile, these observations suggest that the two sites were base paired in the starting WT sequence. Our prior work assumed that such observed compensation would arise with negligible probability if the mutated nucleotides were not paired." This reader was completely confused by this statement. What is different from the assumption in the prior work? I think the meaning got lost in the double negative describing the prior work, but I cannot be sure what he means.

We have eliminated the sentence with the double negative, which was indeed convoluted. To make this abstract statement more concrete, this Results section now presents actual data from our prior study that illustrate cases of complete rescue, no rescue, and partial rescue. Perhaps most importantly, as also suggested by reviewer #2, we include a more detailed cartoon of what is happening to the structural ensemble upon mutations and compensatory mutations in Figure 2.

4) Subsection “Inferring helix frequencies through quantitative compensatory rescue”. Please clearly define exactly what is meant by "helix frequency" in the text. On what experimental metric is it deduced and how is it calculated? If it is in the Materials and methods, then refer to that in the Results section "Helix frequency is defined as XXX and it was calculated as described in Materials and methods".

We have added a definition of helix frequency (“Each data point represents a helix whose rescue factor and simulated helix frequency has been averaged across all its base-pairs.”). We also motivate use of the helix frequency better through a new first Results section explaining why we want to look at helix-helix correlations and what that means in terms of helix frequencies as our desired experimental observable. This early section also includes cartoons of the potential structural ensembles (Figure 1), which helps clarify why helix frequencies are the relevant observables.

5) "[…] median values of the helix frequency posterior probability distributions". What is the meaning of "posterior" in this context? Helix frequency distribution has not even been clearly defined yet – so it is vexing that there is now a posterior form of it.

We neglected to explain terms like posterior distribution, which are used in statistics and Bayesian analysis, but will not be widely known to our target readership. In Results, “These posterior probability distributions represent our belief, informed by experiment, at all helix frequencies from 0% to 100% and provide a complete representation of our uncertainty; ‘fatter’ distributions correspond to larger uncertainties in helix frequencies.” Perhaps most importantly, we now present these distributions over helix frequency as probability density distributions (rather than cumulative probability distributions that rise from 0 to 1, as in our original manuscript) e.g. Figure 3 and 4. These pictures are more intuitive than the previous cumulative distributions – the peaks in these probability density distributions correspond to the most probable values of the helix frequency and the widths convey uncertainty. In presentations to colleagues here at Stanford, these new pictures have been immediately understandable.

6) Subsection “The add riboswitch system”, first two paragraphs. In the Results, the authors finally articulate the structural issues that are being examined in the paper. Almost the entire content of those paragraphs should be used as the Introduction, rather than the present musing on allostery. For example, the reader needs to learn that "The major uncertainty regards what happens when adenine is not present but the RNA nevertheless samples a secondary structure poised to bind adenine, with P1, P2, and P3 formed. This state is termed apoA." It would be clearer to emphasize this right from the beginning, rather than initiating the paper with a treatise on allostery.

We have moved most of this section to the Introduction.

7) "Unexpectedly, in between these two regions, nucleotides 105-111 also modestly increased in SHAPE reactivity upon adenine binding, although, in both the MWC and non-MWC models, this segment is predicted to be sequestered into a P5 helix in all states (Figure 2)." This sounds really interesting, but if one looks closely at the figure and lines up nucleotides 105-111, the effect being referred to is small, and potentially attributable to partial stacking of this region near the helix rather than helical formation itself. This reviewer is concerned that the authors are overinterpreting the small effects that are side-effects of the SHAPE method itself (see point 8).8) A general criticism of the science rather than the writing: A potential flaw in the analysis is the strict interpretation of SHAPE signals in this analysis: The authors assume that shape data can be interpreted in a strictly binary way – representing a population of paired or unpaired duplexes. But SHAPE reactivity is not strictly indicative of base pairing and instead reflects the relative "floppiness" of a 2'-OH – not whether it is paired or not. As a result, there can be partially ordered conformations, and conformationally restricted conformations of a motif that is not double stranded yet can give SHAPE protections of varying magnitude. These issues raise concern about the focus on subtle effects being studied by the investigators.

We fully agree with the reviewer that the subtleties in interpreting SHAPE data preclude use of those data to model structures and structural ensembles, and have published extensively on this information gap (see, e.g., Kladwang et al., “Understanding the Errors of SHAPE-Directed RNA Structure Modeling”, Biochemistry2011 and Tian et al., “High-throughput mutate-map-rescue evaluates SHAPE-directed RNA structure and uncovers excited states”, *RNA* 2014). Indeed, those problems are what motivated us to integrate several layers of mutations with SHAPE to get information that does not rely on specific models for how SHAPE relates to structure or on specific computational models of RNA structure. However, we did not explain that motivation or those caveats well. In the Results section describing estimation of helix frequency from quantitative compensatory rescue, we now start by summarizing a prior study that actually falsified a model guided by SHAPE data alone.

In describing SHAPE reactivity changes that occur upon locking, we now give a disclaimer: “This observation already suggested that mutation-based locking of P1 or P2 opens the gene expression platform, favoring the MWC model over the non-MWC model, although it depends on an assumption that increase in SHAPE reactivity correlates with reduced base pairing, which does not always hold.”.

We have also added an explanation to the Appendix: “Overall, while consistent with available models, these SHAPE data do not discriminate between them due to difficulties in interpreting these data: changes in SHAPE reactivity do not necessarily reflect changes in base pairing frequencies. […] For these reasons, we turned to mutate-and-map measurements (described next) and higher-dimensional techniques (compensatory rescue; described in the main text) that do not require detailed interpretations of the absolute reactivity values at particular nucleotides, but rather changes in reactivities integrated across all nucleotides.”

We also underscore the independence of our new method from the (still poorly understood) relationship of SHAPE data to underlying structure in the first paragraph of Discussion: “Importantly, quantitative inferences of helix-helix correlations from this lockmutate-map-rescue (LM^2^R) workflow do not require interpretation of how SHAPE reactivity corresponds to Watson-Crick pairing probability, which can lead to erroneous conclusions. In that sense, the method is analogous to three-dimensional and four-dimensional pulse sequences that enable structural conclusions to be derived from NMR even as the relationships of chemical shifts to macromolecules structure are not yet understood in quantitative detail.”.

9) Another result that sounds really interesting but is difficult to discern from the data presented: "Compared to the WT sequence, many of these mutants gave protections or enhancements in the 5 ´-most aptamer region of the riboswitch (nts 25-32) and concomitant exposures or protections (respectively) near the 3 ´ end of the riboswitch (nts 105- 122), which included both the start codon and Shine-Dalgarno ribosome binding sequence (arrows, Figure 3)." Exactly where in the figures is this shown? It must be indicated on the display item.

In response to reviewer 2, we have moved these ‘preliminary’ mutate-and-map data to figure supplements and text, and removed this particular statement. Although the noted protections/enhancements in the mutate-and-map data beyond experimental error and are reproducible (e.g., in different salt conditions tested in Cordero and Das, 2015 “Rich RNA structure landscapes revealed by mutate-and-map analysis”), these fine details of SHAPE data are difficult to interpret.

10) Figure 3–Figure 5: Please define "cumulative posterior possibility" in the Results portion of the text (or better yet, come up with a term that is simpler), as an understanding of the parameter is a prerequisite for analyzing the data presented.

Please see response to suggestion 5.

11) Figure 5 and throughout text: If the MWC model is correct, then why isn't the riboswitch constitutively ON 40% of the time in the absence of adenine? Or is the level of "leaky" expression commensurate with the authors' estimates of the population of this state?

The riboswitch is indeed constitutively on at 30-40% level in all prior experiments that have looked at the RNA, even in vivoin *E. coli*. The switch appears leaky. We now state this explicitly in the Discussion: “Functional measurements in vitro and in vivo, use of fluorescent reporters, and single-molecule force experiments all have suggested that the riboswitch is rather ‘leaky’, forming the aptamer secondary structure at ~40% frequency at 20–30 °C, even without adenine.”.

12) "Similarly, compensatory rescue targeting P2 gave a significant helix frequency (42%) when P1 was locked (Figure 4, Figure 4—figure supplement 2, and Table 1). As a further check, both models under question predicted the absence of P4A in apoA; indeed, in either the lock-P1 or lock-P2 background, we did not detect compensatory rescue targeting P4A, and could set an upper bound on its frequency of 4%. Finally, we tested the helix frequency of P4B, which was predicted to be absent (0%) in apoA by the MWC model and present (100%) in the non-MWC model (Figure 2)." This is interesting, but the authors do not indicate exactly which parts of Figure 4 support this model. They simply refer to Figure 4, which is a complex figure lacking any direction to the reader. One cannot evaluate the veracity of the authors' claim.13) Last paragraph of the Results: Again, the authors need to refer directly to the features in Figure 4 that support the helix frequencies being described in the text, and presumably graphed in parts E and F. When inspected by this reviewer, the display items in Figure 4 did not clearly indicate the results claimed. They need to be indicated on the display and referred to in the text.

Figure 4 was too complex because we were showing too much information within single panels and not clearly showing the relevant comparisons. We now show panels, one-by-one for each comparison being made. For example, in Figure 6, we show how the helix frequency of P4B is unambiguously lower when helix P1 is locked by only showing the relevant two traces in one panel, and with a big arrow marked ‘lock’. We also call out these separate figure panels in the main text as they are being described.

14) First paragraph of the Discussion: "Four-dimensional chemical mapping offers an experimental route to this information that is unique in its high throughput and use of standard equipment (capilly electrophoretic sequencers)." It is clear from the data provided in this paper that this is not a high throughput method, in fact it is quite the contrary. Also: one presumes that the authors mean "capillary". Any reference to this being a simple or high throughput experiment should be removed, as it unfairly inflates expectations within the community of the methods that should be reasonably applied in structural studies any other RNA molecule.

We have removed any reference to the method as simple or high-throughput. We fixed the misspelling of “capillary”. We have added two full paragraphs of the Discussion better explaining that the method does indeed require numerous measurements, that there may be limitations for RNAs with longer lengths, and that the method is limited to inferring correlations between helical elements and cannot yet probe correlations involving single-stranded elements or tertiary contacts except insofar as nearby helix elements can act as proxies.

Reviewer #4:[…] 1) At many times the paper is very difficult to follow. Figures are often not described in a way that the reader can access. The language is at times highly specialized and ill defined. If the authors can explain their data in an accessible way, it will greatly increase the impact of the findings.

This concern was also presented by reviewers #2 and #3, and we have made extensive revisions to the presentation as noted above.

2) While an elegant method, 4D chemical mapping is not without its concerns. Our only major concern is the general assumption that locking a helix informs on the natural RNA conformational switch. Could locking a helix prevent certain natural pathways that involve transient interactions between helices, such as pseudoknots involving helices themselves, including defects, or could a lock prevent the need to transiently unfold a helix during the switching.

The presented method is applicable to systems at equilibrium. This regime does appear relevant for molecules like the *add* riboswitch which reversibly turns on and off gene expression through release or hiding of the ribosome binding site; equilibrium measurements and descriptions are also the foundation for our understanding of protein allostery in, e.g., hemoglobin. We do not yet know if the presented 4D chemical mapping might allow dissection of kinetic mechanisms. This limitation was not clear in the original manuscript, and so we have added to Discussion: “As a final caveat, the current LM^2^R method relies on the RNA system being at equilibrium so that the correlation equations (1)-(3) are valid and to convert observed rescue factors to helix frequencies. It will be important to evaluate whether LM^2^R, or some time-dependent extension of the method, could be applied to RNAs that change their structures in highly non-equilibrium scenarios involving co-transcriptional RNA folding and recruitment of energy-dissipating machines like the ribosome or RNA polymerase.”